# PASS-GLM: polynomial approximate sufficient statistics for scalable Bayesian GLM inference

**Jonathan H. Huggins**
CSAIL, MIT
jhuggins@mit.edu

**Ryan P. Adams**
Google Brain and Princeton
rpa@princeton.edu

**Tamara Broderick**
CSAIL, MIT
tbroderick@csail.mit.edu

## Abstract

Generalized linear models (GLMs)—such as logistic regression, Poisson regression, and robust regression—provide interpretable models for diverse data types. Probabilistic approaches, particularly Bayesian ones, allow coherent estimates of uncertainty, incorporation of prior information, and sharing of power across experiments via hierarchical models. In practice, however, the approximate Bayesian methods necessary for inference have either failed to scale to large data sets or failed to provide theoretical guarantees on the quality of inference. We propose a new approach based on constructing polynomial approximate sufficient statistics for GLMs (PASS-GLM). We demonstrate that our method admits a simple algorithm as well as trivial streaming and distributed extensions that do not compound error across computations. We provide theoretical guarantees on the quality of point (MAP) estimates, the approximate posterior, and posterior mean and uncertainty estimates. We validate our approach empirically in the case of logistic regression using a quadratic approximation and show competitive performance with stochastic gradient descent, MCMC, and the Laplace approximation in terms of speed and multiple measures of accuracy—including on an advertising data set with 40 million data points and 20,000 covariates.

## 1 Introduction

Scientists, engineers, and companies increasingly use large-scale data—often only available via streaming—to obtain insights into their respective problems. For instance, scientists might be interested in understanding how varying experimental inputs leads to different experimental outputs; or medical professionals might be interested in understanding which elements of patient histories lead to certain health outcomes. *Generalized linear models* (GLMs) enable these practitioners to explicitly and interpretably model the effect of covariates on outcomes while allowing flexible noise distributions—including binary, count-based, and heavy-tailed observations. Bayesian approaches further facilitate (1) understanding the importance of covariates via coherent estimates of parameter uncertainty, (2) incorporating prior knowledge into the analysis, and (3) sharing of power across different experiments or domains via hierarchical modeling. In practice, however, an exact Bayesian analysis is computationally infeasible for GLMs, so an approximation is necessary. While some approximate methods provide asymptotic guarantees on quality, these methods often only run successfully in the small-scale data regime. In order to run on (at least) millions of data points and thousands of covariates, practitioners often turn to heuristics with no theoretical guarantees on quality. In this work, we propose a novel and simple approximation framework for probabilistic inference in GLMs. We demonstrate theoretical guarantees on the quality of point estimates in the finite-sample setting and on the quality of Bayesian posterior approximations produced by our framework. We show that our framework trivially extends to streaming data and to distributed architectures, with *no* additional compounding of error in these settings. We empirically demonstrate the practicality

of our framework on datasets with up to tens of millions of data points and tens of thousands of covariates.

**Large-scale Bayesian inference.** Calculating accurate approximate Bayesian posteriors for large data sets together with complex models and potentially high-dimensional parameter spaces is a long-standing problem. We seek a method that satisfies the following criteria: (1) it provides a *posterior approximation*; (2) it is *scalable*; (3) it comes equipped with *theoretical guarantees*; and (4) it provides *arbitrarily good approximations*. By *posterior approximation* we mean that the method outputs an approximate posterior distribution, not just a point estimate. By *scalable* we mean that the method examines each data point only a small number of times, and further can be applied to streaming and distributed data. By *theoretical guarantees* we mean that the posterior approximation is certified to be close to the true posterior in terms of, for example, some metric on probability measures. Moreover, the distance between the exact and approximate posteriors is an efficiently computable quantity. By an *arbitrarily good approximation* we mean that, with a large enough computational budget, the method can output an approximation that is as close to the exact posterior as we wish.

Markov chain Monte Carlo (MCMC) methods provide an approximate posterior, and the approximation typically becomes arbitrarily good as the amount of computation time grows asymptotically; thereby MCMC satisfies criteria 1, 3, and 4. But scalability of MCMC can be an issue. Conversely, variational Bayes (VB) and expectation propagation (EP) [27] have grown in popularity due to their scalability to large data and models—though they typically lack guarantees on quality (criteria 3 and 4). Subsampling methods have been proposed to speed up MCMC [1, 5, 6, 21, 25, 41] and VB [18]. Only a few of these algorithms preserve guarantees asymptotic in time (criterion 4), and they often require restrictive assumptions. On the scalability front (criterion 2), many though not all subsampling MCMC methods have been found to require examining a constant fraction of the data at each iteration [2, 6, 7, 30, 31, 38], so the computational gains are limited. Moreover, the random data access required by these methods may be infeasible for very large datasets that do not fit into memory. Finally, they do not apply to streaming and distributed data, and thus fail criterion 2 above. More recently, authors have proposed subsampling methods based on piecewise deterministic Markov processes (PDMPs) [8, 9, 29]. These methods are promising since subsampling data here does not change the invariant distribution of the continuous-time Markov process. But these methods have not yet been validated on large datasets nor is it understood how subsampling affects the mixing rates of the Markov processes. Authors have also proposed methods for coalescing information across distributed computation (criterion 2) in MCMC [12, 32, 34, 35], VB [10, 11], and EP [15, 17]—and in the case of VB, across epochs as streaming data is collected [10, 11]. (See Angelino et al. [3] for a broader discussion of issues surrounding scalable Bayesian inference.) While these methods lead to gains in computational efficiency, they lack rigorous justification and provide no guarantees on the quality of inference (criteria 3 and 4).

To address these difficulties, we are inspired in part by the observation that not all Bayesian models require expensive posterior approximation. When the likelihood belongs to an *exponential family*, Bayesian posterior computation is fast and easy. In particular, it suffices to find the *sufficient statistics* of the data, which require computing a simple summary at each data point and adding these summaries across data points. The latter addition requires a single pass through the data and is trivially streaming or distributed. With the sufficient statistics in hand, the posterior can then be calculated via, e.g., MCMC, and point estimates such as the MLE can be computed—all in time independent of the data set size. Unfortunately, sufficient statistics are not generally available (except in very special cases) for GLMs. We propose to instead develop a notion of *approximate sufficient statistics*. Previously authors have suggested using a *coreset*—a weighted data subset—as a summary of the data [4, 13, 14, 16, 19, 24]. While these methods provide theoretical guarantees on the quality of inference via the model evidence, the resulting guarantees are better suited to approximate optimization and do not translate to guarantees on typical Bayesian desiderata, such as the accuracy of posterior mean and uncertainty estimates. Moreover, while these methods do admit streaming and distributed constructions, the approximation error is compounded across computations.

**Our contributions.** In the present work we instead propose to construct our approximate sufficient statistics via a much simpler *polynomial approximation* for generalized linear models. We therefore call our method *polynomial approximate sufficient statistics for generalized linear models* (PASS-GLM). PASS-GLM satisfies all of the criteria laid of above. It provides a *posterior approximation* with *theoretical guarantees* (criteria 1 and 3). It is *scalable* since is requires only a single pass over

the data and can be applied to streaming and distributed data (criterion 2). And by increasing the number of approximate sufficient statistics, PASS-GLM can produce *arbitrarily good approximations* to the posterior (criterion 4).

The Laplace approximation [39] and variational methods with a Gaussian approximation family [20, 22] may be seen as polynomial (quadratic) approximations in the log-likelihood space. But we note that the VB variants still suffer the issues described above. A Laplace approximation relies on a Taylor series expansion of the log-likelihood around the *maximum a posteriori* (MAP) solution, which requires first calculating the MAP—an expensive multi-pass optimization in the large-scale data setting. Neither Laplace nor VB offers the simplicity of sufficient statistics, including in streaming and distributed computations. The recent work of Stephanou et al. [36] is similar in spirit to ours, though they address a different statistical problem: they construct sequential quantile estimates using Hermite polynomials.

In the remainder of the paper, we begin by describing generalized linear models in more detail in Section 2. We construct our novel polynomial approximation and specify our PASS-GLM algorithm in Section 3. We will see that streaming and distributed computation are trivial for our algorithm and do not compound error. In Section 4.1, we demonstrate finite-sample guarantees on the quality of the MAP estimate arising from our algorithm, with the maximum likelihood estimate (MLE) as a special case. In Section 4.2, we prove guarantees on the Wasserstein distance between the exact and approximate posteriors—and thereby bound both posterior-derived point estimates and uncertainty estimates. In Section 5, we demonstrate the efficacy of our approach in practice by focusing on logistic regression. We demonstrate experimentally that PASS-GLM can be scaled with almost no loss of efficiency to multi-core architectures. We show on a number of real-world datasets—including a large, high-dimensional advertising dataset (40 million examples with 20,000 dimensions)—that PASS-GLM provides an attractive trade-off between computation and accuracy.

## 2 Background

**Generalized linear models.** *Generalized linear models* (GLMs) combine the interpretability of linear models with the flexibility of more general outcome distributions—including binary, ordinal, and heavy-tailed observations. Formally, we let $\mathcal{Y} \subseteq \mathbb{R}$ be the observation space, $\mathcal{X} \subseteq \mathbb{R}^d$ be the covariate space, and $\Theta \subseteq \mathbb{R}^d$ be the parameter space. Let $\mathcal{D} := \{(\mathbf{x}_n, y_n)\}_{n=1}^N$ be the observed data. We write $\mathbf{X} \in \mathbb{R}^{N \times d}$ for the matrix of all covariates and $\mathbf{y} \in \mathbb{R}^N$ for the vector of all observations. We consider GLMs

$$\log p(\mathbf{y} \mid \mathbf{X}, \boldsymbol{\theta}) = \sum_{n=1}^N \log p(y_n \mid g^{-1}(\mathbf{x}_n \cdot \boldsymbol{\theta})) = \sum_{n=1}^N \phi(y_n, \mathbf{x}_n \cdot \boldsymbol{\theta}),$$

where $\mu := g^{-1}(\mathbf{x}_n \cdot \boldsymbol{\theta})$ is the expected value of $y_n$ and $g^{-1} : \mathbb{R} \to \mathbb{R}$ is the *inverse link function*. We call $\phi(y, s) := \log p(y \mid g^{-1}(s))$ the GLM *mapping function*.

Examples include some of the most widely used models in the statistical toolbox. For instance, for binary observations $y \in \{\pm 1\}$, the likelihood model is Bernoulli, $p(y = 1 \mid \mu) = \mu$, and the link function is often either the logit $g(\mu) = \log \frac{\mu}{1-\mu}$ (as in logistic regression) or the probit $g(\mu) = \Phi^{-1}(\mu)$, where $\Phi$ is the standard Gaussian CDF. When modeling count data $y \in \mathbb{N}$, the likelihood model might be Poisson, $p(y \mid \mu) = \mu^y e^{-\mu}/y!$, and $g(\mu) = \log(\mu)$ is the typical log link. Other GLMs include gamma regression, robust regression, and binomial regression, all of which are commonly used for large-scale data analysis (see Examples A.1 and A.3).

If we place a prior $\pi_0(\mathrm{d}\boldsymbol{\theta})$ on the parameters, then a full Bayesian analysis aims to approximate the (typically intractable) GLM posterior distribution $\pi_{\mathcal{D}}(\mathrm{d}\boldsymbol{\theta})$, where

$$\pi_{\mathcal{D}}(\mathrm{d}\boldsymbol{\theta}) = \frac{p(\mathbf{y} \mid \mathbf{X}, \boldsymbol{\theta}) \, \pi_0(\mathrm{d}\boldsymbol{\theta})}{\int p(\mathbf{y} \mid \mathbf{X}, \boldsymbol{\theta}') \, \pi_0(\mathrm{d}\boldsymbol{\theta}')}.$$

The *maximum a posteriori* (MAP) solution gives a point estimate of the parameter:

$$\boldsymbol{\theta}_{\mathrm{MAP}} := \operatorname*{arg\,max}_{\boldsymbol{\theta} \in \Theta} \pi_{\mathcal{D}}(\boldsymbol{\theta}) = \operatorname*{arg\,max}_{\boldsymbol{\theta} \in \Theta} \log \pi_0(\boldsymbol{\theta}) + \mathcal{L}_{\mathcal{D}}(\boldsymbol{\theta}), \tag{1}$$

where $\mathcal{L}_{\mathcal{D}}(\boldsymbol{\theta}) := \log p(\mathbf{y} \mid \mathbf{X}, \boldsymbol{\theta})$ is the data log-likelihood. The MAP problem strictly generalizes finding the maximum likelihood estimate (MLE), since the MAP solution equals the MLE when using the (possibly improper) prior $\pi_0(\boldsymbol{\theta}) = 1$.

---

**Algorithm 1** PASS-GLM inference

---

**Require:** data $\mathcal{D}$, GLM mapping function $\phi : \mathbb{R} \to \mathbb{R}$, degree $M$, polynomial basis $(\psi_m)_{m \in \mathbb{N}}$ with base measure $\varsigma$

1: Calculate basis coefficients $b_m \leftarrow \int \phi \psi_m \mathrm{d}\varsigma$ using numerical integration for $m = 0, \dots, M$
2: Calculate polynomial coefficients $b_m^{(M)} \leftarrow \sum_{k=m}^{M} \alpha_{k,m} b_m$ for $m = 0, \dots, M$
3: **for** $\mathbf{k} \in \mathbb{N}^d$ with $\sum_j k_j \le M$ **do**
4:     Initialize $t_{\mathbf{k}} \leftarrow 0$
5: **for** $n = 1, \dots, N$ **do**         ▷ Can be done with any combination of batch, parallel, or streaming
6:     **for** $\mathbf{k} \in \mathbb{N}^d$ with $\sum_j k_j \le M$ **do**
7:         Update $t_{\mathbf{k}} \leftarrow t_{\mathbf{k}} + (y_n \mathbf{x}_n)^{\mathbf{k}}$
8: Form approximate log-likelihood $\tilde{\mathcal{L}}_{\mathcal{D}}(\boldsymbol{\theta}) = \sum_{\mathbf{k} \in \mathbb{N}^d : \sum_j k_j \le m} \binom{m}{\mathbf{k}} b_m^{(M)} t_{\mathbf{k}} \boldsymbol{\theta}^{\mathbf{k}}$
9: Use $\tilde{\mathcal{L}}_{\mathcal{D}}(\boldsymbol{\theta})$ to construct approximate posterior $\tilde{\pi}_{\mathcal{D}}(\boldsymbol{\theta})$

---

**Computation and exponential families.** In large part due to the high-dimensional integral implicit in the normalizing constant, approximating the posterior, e.g., via MCMC or VB, is often prohibitively expensive. Approximating this integral will typically require many evaluations of the (log-)likelihood, or its gradient, and each evaluation may require $\Omega(N)$ time.

Computation is much more efficient, though, if the model is in an *exponential family* (EF). In the EF case, there exist functions $\mathbf{t}, \boldsymbol{\eta} : \mathbb{R}^d \to \mathbb{R}^m$, such that[1]

$$\log p(y_n \mid \mathbf{x}_n, \boldsymbol{\theta}) = \mathbf{t}(y_n, \mathbf{x}_n) \cdot \boldsymbol{\eta}(\boldsymbol{\theta}) =: \mathcal{L}_{\mathcal{D},\text{EF}}(\boldsymbol{\theta}; \mathbf{t}(y_n, \mathbf{x}_n)).$$

Thus, we can rewrite the log-likelihood as

$$\mathcal{L}_{\mathcal{D}}(\boldsymbol{\theta}) = \sum_{n=1}^{N} \mathcal{L}_{\mathcal{D},\text{EF}}(\boldsymbol{\theta}; \mathbf{t}(y_n, \mathbf{x}_n)) =: \mathcal{L}_{\mathcal{D},\text{EF}}(\boldsymbol{\theta}; \mathbf{t}(\mathcal{D})),$$

where $\mathbf{t}(\mathcal{D}) := \sum_{n=1}^{N} \mathbf{t}(y_n, \mathbf{x}_n)$. The *sufficient statistics* $\mathbf{t}(\mathcal{D})$ can be calculated in $O(N)$ time, after which each evaluation of $\mathcal{L}_{\mathcal{D},\text{EF}}(\boldsymbol{\theta}; \mathbf{t}(\mathcal{D}))$ or $\nabla \mathcal{L}_{\mathcal{D},\text{EF}}(\boldsymbol{\theta}; \mathbf{t}(\mathcal{D}))$ requires only $O(1)$ time. Thus, instead of $K$ passes over $N$ data (requiring $O(NK)$ time), only $O(N + K)$ time is needed. Even for moderate values of $N$, the time savings can be substantial when $K$ is large.

The Poisson distribution is an illustrative example of a one-parameter exponential family with $\mathbf{t}(y) = (1, y, \log y!)$ and $\boldsymbol{\eta}(\theta) = (\theta, \log \theta, 1)$. Thus, if we have data $\mathbf{y}$ (there are no covariates), $\mathbf{t}(\mathbf{y}) = (N, \sum_n y_n, \sum \log y_n!)$. In this case it is easy to calculate that the maximum likelihood estimate of $\theta$ from $\mathbf{t}(\mathbf{y})$ as $t_1(\mathbf{y})/t_0(\mathbf{y}) = N^{-1} \sum_n y_n$.

Unfortunately, GLMs rarely belong to an exponential family – even if the outcome distribution is in an exponential family, the use of a link destroys the EF structure. In logistic regression, we write (overloading the $\phi$ notation) $\log p(y_n \mid \mathbf{x}_n, \boldsymbol{\theta}) = \phi_{\text{logit}}(y_n \mathbf{x}_n \cdot \boldsymbol{\theta})$, where $\phi_{\text{logit}}(s) := -\log(1 + e^{-s})$. For Poisson regression with log link, $\log p(y_n \mid \mathbf{x}_n, \boldsymbol{\theta}) = \phi_{\text{Poisson}}(y_n, \mathbf{x}_n \cdot \boldsymbol{\theta})$, where $\phi_{\text{Poisson}}(y, s) := ys - e^s - \log y!$. In both cases, we cannot express the log-likelihood as an inner product between a function solely of the data and a function solely of the parameter.

## 3 PASS-GLM

Since exact sufficient statistics are not available for GLMs, we propose to construct *approximate sufficient statistics*. In particular, we propose to approximate the mapping function $\phi$ with an order-$M$ polynomial $\phi_M$. We therefore call our method *polynomial approximate sufficient statistics for GLMs* (PASS-GLM). We illustrate our method next in the logistic regression case, where $\log p(y_n \mid \mathbf{x}_n, \boldsymbol{\theta}) = \phi_{\text{logit}}(y_n \mathbf{x}_n \cdot \boldsymbol{\theta})$. The fully general treatment appears in Appendix A. Let $b_0^{(M)}, b_1^{(M)} \dots, b_M^{(M)}$ be constants such that

$$\phi_{\text{logit}}(s) \approx \phi_M(s) := \sum_{m=0}^{M} b_m^{(M)} s^m.$$

Let $\mathbf{v}^{\mathbf{k}} := \prod_{j=1}^{d} v_j^{k_j}$ for vectors $\mathbf{v}, \mathbf{k} \in \mathbb{R}^d$. Taking $s = y\mathbf{x} \cdot \boldsymbol{\theta}$, we obtain

$$\phi_{\mathsf{logit}}(y\mathbf{x} \cdot \boldsymbol{\theta}) \approx \phi_M(y\mathbf{x} \cdot \boldsymbol{\theta}) = \sum_{m=0}^{M} b_m^{(M)}(y\mathbf{x} \cdot \boldsymbol{\theta})^m = \sum_{m=0}^{M} b_m^{(M)} \sum_{\substack{\mathbf{k} \in \mathbb{N}^d \\ \sum_j k_j = m}} \binom{m}{\mathbf{k}}(y\mathbf{x})^{\mathbf{k}}\boldsymbol{\theta}^{\mathbf{k}}$$

$$= \sum_{m=0}^{M} \sum_{\mathbf{k} \in \mathbb{N}^d : \sum_j k_j = m} a(\mathbf{k}, m, M)(y\mathbf{x})^{\mathbf{k}}\boldsymbol{\theta}^{\mathbf{k}},$$

where $\binom{m}{\mathbf{k}}$ is the multinomial coefficient and $a(\mathbf{k}, m, M) := \binom{m}{\mathbf{k}}b_m^{(M)}$. Thus, $\phi_M$ is an $M$-degree polynomial approximation to $\phi_{\mathsf{logit}}(y\mathbf{x} \cdot \boldsymbol{\theta})$ with the $\binom{d+M}{d}$ monomials of degree at most $M$ serving as sufficient statistics derived from $y\mathbf{x}$. Specifically, we have a exponential family model with

$$\mathbf{t}(y\mathbf{x}) = ([y\mathbf{x}]^{\mathbf{k}})_{\mathbf{k}} \qquad \text{and} \qquad \boldsymbol{\eta}(\boldsymbol{\theta}) = (a(\mathbf{k}, m, M)\boldsymbol{\theta}^{\mathbf{k}})_{\mathbf{k}},$$

where $\mathbf{k}$ is taken over all $\mathbf{k} \in \mathbb{N}^d$ such that $\sum_j k_j \leq M$. We next discuss the calculation of the $b_m^{(M)}$ and the choice of $M$.

**Choosing the polynomial approximation.** To calculate the coefficients $b_m^{(M)}$, we choose a polynomial basis $(\psi_m)_{m \in \mathbb{N}}$ orthogonal with respect to a base measure $\varsigma$, where $\psi_m$ is degree $m$ [37]. That is, $\psi_m(s) = \sum_{j=0}^{m} \alpha_{m,j}s^j$ for some $\alpha_{m,j}$, and $\int \psi_m \psi_{m'} \mathrm{d}\varsigma = \delta_{mm'}$, where $\delta_{mm'} = 1$ if $m = m'$ and zero otherwise. If $b_m := \int \phi\psi_m \mathrm{d}\varsigma$, then $\phi(s) = \sum_{m=0}^{\infty} b_m\psi_m(s)$ and the approximation $\phi_M(s) = \sum_{m=0}^{M} b_m\psi_m(s)$. Conclude that $b_m^{(M)} = \sum_{k=m}^{M} \alpha_{k,m}b_k$. The complete PASS-GLM framework appears in Algorithm 1.

Choices for the orthogonal polynomial basis include Chebyshev, Hermite, Leguerre, and Legendre polynomials [37]. We choose Chebyshev polynomials since they provide a uniform quality guarantee on a finite interval, e.g., $[-R, R]$ for some $R > 0$ in what follows. If $\phi$ is smooth, the choice of Chebyshev polynomials (scaled appropriately, along with the base measure $\varsigma$, based on the choice of $R$) yields error exponentially small in $M$: $\sup_{s \in [-R, R]} |\phi(s) - \phi_M(s)| \leq C\rho^M$ for some $0 < \rho < 1$ and $C > 0$ [26]. We show in Appendix B that the error in the approximate derivative $\phi'_M$ is also exponentially small in $M$: $\sup_{s \in [-R, R]} |\phi'(s) - \phi'_M(s)| \leq C'\rho^M$, where $C' > C$.

**Choosing the polynomial degree.** For fixed $d$, the number of monomials is $O(M^d)$ while for fixed $M$ the number of monomials is $O(d^M)$. The number of approximate sufficient statistics can remain manageable when either $M$ or $d$ is small but becomes unwieldy if $M$ and $d$ are both large. Since our experiments (Section 5) generally have large $d$, we focus on the small $M$ case here.

In our experiments we further focus on the choice of logistic regression as a particularly popular GLM example with $p(y_n \mid \mathbf{x}_n, \boldsymbol{\theta}) = \phi_{\mathsf{logit}}(y_n\mathbf{x}_n \cdot \boldsymbol{\theta})$, where $\phi_{\mathsf{logit}}(s) := -\log(1 + e^{-s})$. In general, the smallest and therefore most compelling choice of $M$ *a priori is 2*, and we demonstrate the reasonableness of this choice empirically in Section 5 for a number of large-scale data analyses. In addition, in the logistic regression case, $M = 6$ is the next usable choice beyond $M = 2$. This is because $b_{2k+1}^{(M)} = 0$ for all integer $k \geq 1$ with $2k + 1 \leq M$. So any approximation beyond $M = 2$ must have $M \geq 4$. Also, $b_{4k}^{(M)} > 0$ for all integers $k \geq 1$ with $4k \leq M$. So choosing $M = 4k$, $k \geq 1$, leads to a pathological approximation of $\phi_{\mathsf{logit}}$ where the log-likelihood can be made arbitrarily large by taking $\|\boldsymbol{\theta}\|_2 \to \infty$. Thus, a reasonable polynomial approximation for logistic regression requires $M = 2 + 4k$, $k \geq 0$. We have discussed the relative drawbacks of other popular quadratic approximations, including the Laplace approximation and variational methods, in Section 1.

# 4 Theoretical Results

We next establish quality guarantees for PASS-GLM. We first provide finite-sample and asymptotic guarantees on the MAP (point estimate) solution, and therefore on the MLE, in Section 4.1. We then provide guarantees on the Wasserstein distance between the approximate and exact posteriors, and show these bounds translate into bounds on the quality of posterior mean and uncertainty estimates, in Section 4.2. See Appendix C for extended results, further discussion, and all proofs.

## 4.1 MAP approximation

In Appendix C, we state and prove Theorem C.1, which provides guarantees on the quality of the MAP estimate for an arbitrary approximation $\tilde{\mathcal{L}}_{\mathcal{D}}(\boldsymbol{\theta})$ to the log-likelihood $\mathcal{L}_{\mathcal{D}}(\boldsymbol{\theta})$. The approximate

MAP (i.e., the MAP under $\tilde{\mathcal{L}}_{\mathcal{D}}$) is (cf. Eq. (1))

$$\tilde{\boldsymbol{\theta}}_{\text{MAP}} := \arg\max_{\boldsymbol{\theta} \in \Theta} \log \pi_0(\boldsymbol{\theta}) + \tilde{\mathcal{L}}_{\mathcal{D}}(\boldsymbol{\theta}).$$

Roughly, we find in Theorem C.1 that the error in the MAP estimate naturally depends on the error of the approximate log-likelihood as well as the peakedness of the posterior near the MAP. In the latter case, if $\log \pi_{\mathcal{D}}$ is very flat, then even a small error from using $\tilde{\mathcal{L}}_{\mathcal{D}}$ in place of $\mathcal{L}_{\mathcal{D}}$ could lead to a large error in the approximate MAP solution. We measure the peakedness of the distribution in terms of the strong convexity constant[2] of $-\log \pi_{\mathcal{D}}$ near $\boldsymbol{\theta}_{\text{MAP}}$.

We apply Theorem C.1 to PASS-GLM for logistic regression and robust regression. We require the assumption that

$$\phi_M(t) \leq \phi(t) \ \forall t \notin [-R, R], \tag{2}$$

which in the cases of logistic regression and smoothed Huber regression, we conjecture holds for $M = 2 + 4k$, $k \in \mathbb{N}$. For a matrix $\mathbf{A}$, $\|\mathbf{A}\|_2$ denotes its spectral norm.

**Corollary 4.1.** *For the logistic regression model, assume that $\|(\nabla^2 \mathcal{L}_{\mathcal{D}}(\boldsymbol{\theta}_{MAP}))^{-1}\|_2 \leq cd/N$ for some constant $c > 0$ and that $\|\mathbf{x}_n\|_2 \leq 1$ for all $n = 1, \ldots, N$. Let $\phi_M$ be the order-$M$ Chebyshev approximation to $\phi_{\text{logit}}$ on $[-R, R]$ such that Eq. (2) holds. Let $\tilde{\pi}_{\mathcal{D}}(\boldsymbol{\theta})$ denote the posterior approximation obtained by using $\phi_M$ with a log-concave prior. Then there exist numbers $r = r(R) > 1$, $\varepsilon = \varepsilon(M) = O(r^{-M})$, and $\alpha^* \geq \frac{27}{\varepsilon d^3 c^3 + 54}$, such that if $R - \|\boldsymbol{\theta}_{MAP}\|_2 \geq 2\sqrt{\frac{cd\varepsilon}{\alpha^*}}$, then*

$$\|\boldsymbol{\theta}_{MAP} - \tilde{\boldsymbol{\theta}}_{MAP}\|_2^2 \leq \frac{4cd\varepsilon}{\alpha^*} \leq \frac{4}{27}c^4 d^4 \varepsilon^2 + 8cd\varepsilon.$$

The main takeaways from Corollary 4.1 are that (1) the error decreases exponentially in $M$ thanks to the $\varepsilon$ term, (2) the error does not depend on the amount of data, and (3) in order for the bound on the approximate MAP solution to hold, the norm of the true MAP solution must be sufficiently smaller than $R$.

*Remark* 4.2. Some intuition for the assumption on the Hessian of $\mathcal{L}_{\mathcal{D}}$, i.e., $\nabla^2 \mathcal{L}_{\mathcal{D}}(\boldsymbol{\theta}) = \sum_{n=1}^N \phi''_{\text{logit}}(y_n \mathbf{x}_n \cdot \boldsymbol{\theta}) \mathbf{x}_n \mathbf{x}_n^\top$, is as follows. Typically for $\boldsymbol{\theta}$ near $\boldsymbol{\theta}_{\text{MAP}}$, the minimum eigenvalue of $\nabla^2 \mathcal{L}_{\mathcal{D}}(\boldsymbol{\theta})$ is at least $N/(cd)$ for some $c > 0$. The minimum eigenvalue condition in Corollary 4.1 holds if, for example, a constant fraction of the data satisfies $0 < b \leq \|x_n\|_2 \leq B < \infty$ and that subset of the data does not lie too close to any $(d-1)$-dimensional hyperplane. This condition essentially requires the data not to be degenerate and is similar to ones used to show asymptotic consistency of logistic regression [40, Ex. 5.40].

The approximate MAP error bound in the robust regression case using, for example, the smoothed Huber loss (Example A.1), is quite similar to the logistic regression result.

**Corollary 4.3.** *For robust regression with smoothed Huber loss, assume that a constant fraction of the data satisfies $|\mathbf{x}_n \cdot \boldsymbol{\theta}_{MAP} - y_n| \leq b/2$ and that $\|\mathbf{x}_n\|_2 \leq 1$ for all $n = 1, \ldots, N$. Let $\phi_M$ be the order $M$ Chebyshev approximation to $\phi_{\text{Huber}}$ on $[-R, R]$ such that Eq. (2) holds. Let $\tilde{\pi}_{\mathcal{D}}(\boldsymbol{\theta})$ denote the posterior approximation obtained by using $\phi_M$ with a log-concave prior. Then if $R \gg \|\boldsymbol{\theta}_{MAP}\|_2$, there exists $r > 1$ such that for $M$ sufficiently large, $\|\boldsymbol{\theta}_{MAP} - \tilde{\boldsymbol{\theta}}_{MAP}\|_2^2 = O(dr^{-M})$.*

## 4.2 Posterior approximation

We next establish guarantees on how close the approximate and exact posteriors are in *Wasserstein distance*, $d_{\mathcal{W}}$. For distributions $P$ and $Q$ on $\mathbb{R}^d$, $d_{\mathcal{W}}(P, Q) := \sup_{f:\|f\|_L \leq 1} |\int f \mathrm{d}P - \int f \mathrm{d}Q|$, where $\|f\|_L$ denotes the Lipschitz constant of $f$.[3] This choice of distance is particularly useful since, if $d_{\mathcal{W}}(\pi_{\mathcal{D}}, \tilde{\pi}_{\mathcal{D}}) \leq \delta$, then $\tilde{\pi}_{\mathcal{D}}$ can be used to estimate any function with bounded gradient with error at most $\delta \sup_{\mathbf{w}} \|\nabla f(\mathbf{w})\|_2$. Wasserstein error bounds therefore give bounds on the mean estimates (corresponding to $f(\boldsymbol{\theta}) = \theta_i$) as well as uncertainty estimates such as mean absolute deviation (corresponding to $f(\boldsymbol{\theta}) = |\bar{\theta}_i - \theta_i|$, where $\bar{\theta}_i$ is the expected value of $\theta_i$).

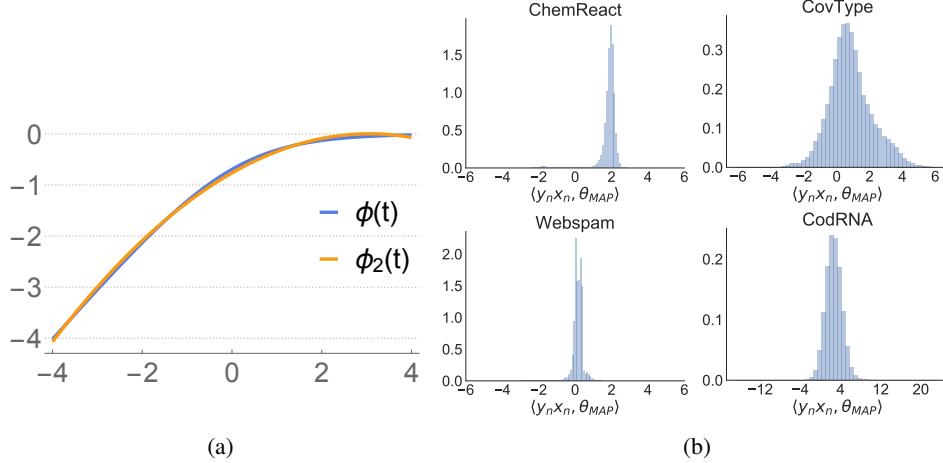

Figure 1: Validating the use of PASS-GLM with $M = 2$. **(a)** The second-order Chebyshev approximation to $\phi = \phi_{\text{logit}}$ on $[-4, 4]$ is very accurate, with error of at most 0.069. **(b)** For a variety of datasets, the inner products $\langle y_n \mathbf{x}_n, \boldsymbol{\theta}_{\text{MAP}} \rangle$ are mostly in the range of $[-4, 4]$.

Our general result (Theorem C.3) is stated and proved in Appendix C. Similar to Theorem C.1, the result primarily depends on the peakedness of the approximate posterior and the error of the approximate gradients. If the gradients are poorly approximated then the error can be large while if the (approximate) posterior is flat then even small gradient errors could lead to large shifts in expected values of the parameters and hence large Wasserstein error.

We apply Theorem C.3 to PASS-GLM for logistic regression and Poisson regression. We give simplified versions of these corollaries in the main text and defer the more detailed versions to Appendix C. For logistic regression we assume $M = 2$ and $\Theta = \mathbb{R}^d$ since this is the setting we use for our experiments. The result is similar in spirit to Corollary 4.1, though more straightforward since $M = 2$. Critically, we see in this result how having small error depends on $|y_n \mathbf{x}_n \cdot \bar{\boldsymbol{\theta}}| \leq R$ with high probability. Otherwise the second term in the bound will be large.

**Corollary 4.4.** *Let $\phi_2$ be the second-order Chebyshev approximation to $\phi_{\text{logit}}$ on $[-R, R]$ and let $\tilde{\pi}_{\mathcal{D}}(\boldsymbol{\theta}) = \mathcal{N}(\boldsymbol{\theta} \mid \tilde{\boldsymbol{\theta}}_{MAP}, \tilde{\boldsymbol{\Sigma}})$ denote the posterior approximation obtained by using $\phi_2$ with a Gaussian prior $\pi_0(\boldsymbol{\theta}) = \mathcal{N}(\boldsymbol{\theta} \mid \boldsymbol{\theta}_0, \boldsymbol{\Sigma}_0)$. Let $\bar{\boldsymbol{\theta}} := \int \boldsymbol{\theta} \pi_{\mathcal{D}}(\mathrm{d}\boldsymbol{\theta})$, let $\delta_1 := N^{-1} \sum_{n=1}^N \langle y_n \mathbf{x}_n, \bar{\boldsymbol{\theta}} \rangle$, and let $\sigma_1$ be the subgaussianity constant of the random variable $\langle y_n \mathbf{x}_n, \bar{\boldsymbol{\theta}} \rangle - \delta_1$, where $n \sim \mathsf{Unif}\{1, \ldots, N\}$. Assume that $|\delta_1| \leq R$, that $\|\tilde{\boldsymbol{\Sigma}}\|_2 \leq cd/N$, and that $\|\mathbf{x}_n\|_2 \leq 1$ for all $n = 1, \ldots, N$. Then with $\sigma_0^2 := \|\boldsymbol{\Sigma}_0\|_2$, we have*

$$d_{\mathcal{W}}(\pi_{\mathcal{D}}, \tilde{\pi}_{\mathcal{D}}) = O\left(dR^4 + d\sigma_0 \exp\left(\sigma_1^2 \sigma_0^{-2} - \sqrt{2}\sigma_0^{-1}(R - |\delta_1|)\right)\right).$$

The main takeaway from Corollary 4.4 is that if (a) for most $n$, $|\langle \mathbf{x}_n, \bar{\boldsymbol{\theta}} \rangle| < R$, so that $\phi_2$ is a good approximation to $\phi_{\text{logit}}$, and (b) the approximate posterior concentrates quickly, then we get a high-quality approximate posterior. This result matches up with the experimental results (see Section 5 for further discussion).

For Poisson regression, we return to the case of general $M$. Recall that in the Poisson regression model that the expectation of $y_n$ is $\mu = e^{\mathbf{x}_n \cdot \boldsymbol{\theta}}$. If $y_n$ is bounded and has non-trivial probability of being greater than zero, we lose little by restricting $\mathbf{x}_n \cdot \boldsymbol{\theta}$ to be bounded. Thus, we will assume that the parameter space is bounded. As in Corollaries 4.1 and 4.3, the error is exponentially small in $M$ and, as long as $\|\sum_{n=1}^N \mathbf{x}_n \mathbf{x}_n^\top\|_2$ grows linearly in $N$, does not depend on the amount of data.

**Corollary 4.5.** *Let $f_M(s)$ be the order-$M$ Chebyshev approximation to $e^t$ on the interval $[-R, R]$, and let $\tilde{\pi}_{\mathcal{D}}(\boldsymbol{\theta})$ denote the posterior approximation obtained by using the approximation $\log \tilde{p}(y_n \mid \mathbf{x}_n, \boldsymbol{\theta}) := y_n \mathbf{x}_n \cdot \boldsymbol{\theta} - f_M(\mathbf{x}_n \cdot \boldsymbol{\theta}) - \log y_n!$ with a log-concave prior on $\Theta = \mathbb{B}_R(\mathbf{0})$. If $\inf_{s \in [-R, R]} f_M''(s) \geq \tilde{\varrho} > 0$, $\|\sum_{n=1}^N \mathbf{x}_n \mathbf{x}_n^\top\|_2 = \Omega(N/d)$, and $\|\mathbf{x}_n\|_2 \leq 1$ for all $n = 1, \ldots, N$, then*

$$d_{\mathcal{W}}(\pi_{\mathcal{D}}, \tilde{\pi}_{\mathcal{D}}) = O\left(d\tilde{\varrho}^{-1} M^2 e^R 2^{-M}\right).$$

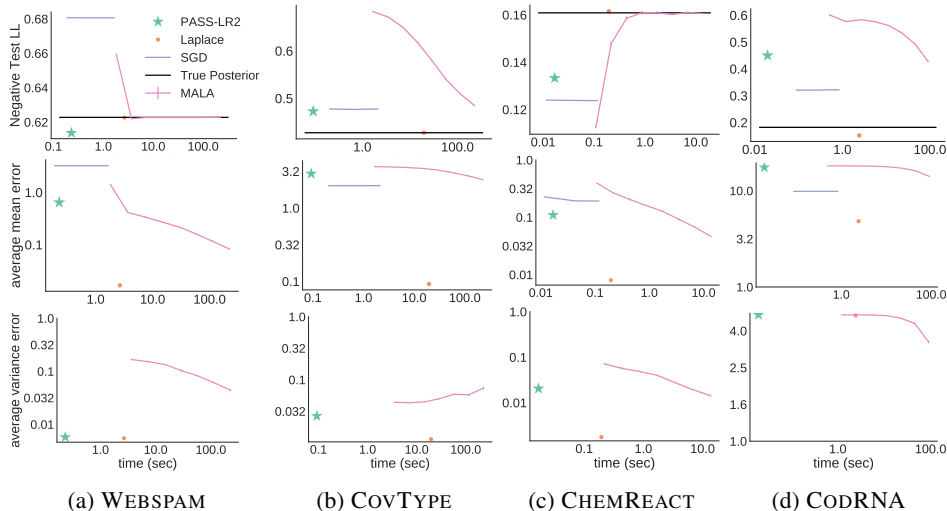

Figure 2: Batch inference results. In all metrics smaller is better.

Note that although $\tilde{\varrho}^{-1}$ does depend on $R$ and $M$, as $M$ becomes large it converges to $e^R$. Observe that if we truncate a prior on $\mathbb{R}^d$ to be on $\mathbb{B}_R(\mathbf{0})$, by making $R$ and $M$ sufficiently large, the Wasserstein distance between $\pi_\mathcal{D}$ and the PASS-GLM posterior approximation $\tilde{\pi}_\mathcal{D}$ can be made arbitarily small. Similar results could be shown for other GLM likelihoods.

## 5 Experiments

In our experiments, we focus on logistic regression, a particularly popular GLM example.[4] As discussed in Section 3, we choose $M = 2$ and call our algorithm PASS-LR2. Empirically, we observe that $M = 2$ offers a high-quality approximation of $\phi$ on the interval $[-4, 4]$ (Fig. 1a). In fact $\sup_{s \in [-4,4]} |\phi_2(s) - \phi(s)| < 0.069$. Moreover, we observe that for many datasets, the inner products $y_n \mathbf{x}_n \cdot \boldsymbol{\theta}_{\text{MAP}}$ tend to be concentrated within $[-4, 4]$, and therefore a high-quality approximation on this range is sufficient for our analysis. In particular, Fig. 1b shows histograms of $y_n \mathbf{x}_n \cdot \boldsymbol{\theta}_{\text{MAP}}$ for four datasets from our experiments. In all but one case, over 98% of the data points satisfy $|y_n \mathbf{x}_n \cdot \boldsymbol{\theta}_{\text{MAP}}| \leq 4$. In the remaining dataset (CODRNA), only $\sim$80% of the data satisfy this condition, and this is the dataset for which PASS-LR2 performed most poorly (cf. Corollary 4.4).

### 5.1 Large dataset experiments

In order to compare PASS-LR2 to other approximate Bayesian methods, we first restrict our attention to datasets with fewer than 1 million data points. We compare to the Laplace approximation and the adaptive Metropolis-adjusted Langevin algorithm (MALA). We also compare to stochastic gradient descent (SGD) although SGD provides only a point estimate and no approximate posterior. In all experiments, no method performs as well as PASS-LR2 given the same (or less) running time.

**Datasets.** The CHEMREACT dataset consists of $N = 26{,}733$ chemicals, each with $d = 100$ properties. The goal is to predict whether each chemical is reactive. The WEBSPAM corpus consists of $N = 350{,}000$ web pages and the covariates consist of the $d = 127$ features that each appear in at least 25 documents. The cover type (COVTYPE) dataset consists of $N = 581{,}012$ cartographic observations with $d = 54$ features. The task is to predict the type of trees that are present at each observation location. The CODRNA dataset consists of $N = 488{,}565$ and $d = 8$ RNA-related features. The task is to predict whether the sequences are non-coding RNA.

Fig. 2 shows average errors of the posterior mean and variance estimates as well as negative test log-likelihood for each method versus the time required to run the method. SGD was run for between 1 and 20 epochs. The true posterior was estimated by running three chains of adaptive MALA for 50,000 iterations, which produced Gelman-Rubin statistics well below 1.1 for all datasets.

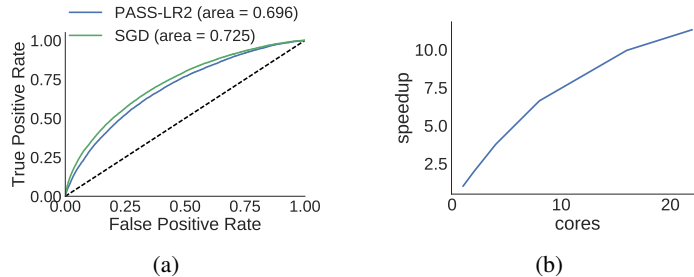

(a)                         (b)

Figure 3: **(a)** ROC curves for streaming inference on 40 million CRITEO data points. SGD and PASS-LR2 had negative test log-likelihoods of, respectively, 0.07 and 0.045. **(b)** Cores vs. speedup (compared to one core) for parallelization experiment on 6 million examples from the CRITEO dataset.

**Speed.** For all four datasets, PASS-LR2 was an order of magnitude faster than SGD and 2–3 orders of magnitude faster than the Laplace approximation. **Mean and variance estimates.** For CHEM-REACT, WEBSPAM, and COVTYPE, PASS-LR2 was superior to or competitive with SGD, with MALA taking 10–100x longer to produce comparable results. Laplace again outperformed all other methods. Critically, on all datasets the PASS-LR2 variance estimates were competitive with Laplace and MALA. **Test log-likelihood.** For CHEMREACT and WEBSPAM, PASS-LR2 produced results competitive with all other methods. MALA took 10–100x longer to produce comparable results. For COVTYPE, PASS-LR2 was competitive with SGD but took a tenth of the time, and MALA took 1000x longer for comparable results. Laplace outperformed all other methods, but was orders of magnitude slower than PASS-LR2. CODRNA was the only dataset where PASS-LR2 performed poorly. However, this performance was expected based on the $y_n \mathbf{x}_n \cdot \boldsymbol{\theta}_{\mathrm{MAP}}$ histogram (Fig. 1a).

## 5.2   Very large dataset experiments using streaming and distributed PASS-GLM

We next test PASS-LR2, which is streaming without requiring any modifications, on a subset of 40 million data points from the Criteo terabyte ad click prediction dataset (CRITEO). The covariates are 13 integer-valued features and 26 categorical features. After one-hot encoding, on the subset of the data we considered, $d \approx 3$ million. For tractability we used sparse random projections [23] to reduce the dimensionality to 20,000. At this scale, comparing to the other fully Bayesian methods from Section 5.1 was infeasible; we compare only to the predictions and point estimates from SGD. PASS-LR2 performs slightly worse than SGD in AUC (Fig. 3a), but outperforms SGD in negative test log-likelihood (0.07 for SGD, 0.045 for PASS-LR2). Since PASS-LR2 estimates a full covariance, it was about 10x slower than SGD. A promising approach to speeding up and reducing memory usage of PASS-LR2 would be to use a low-rank approximation to the second-order moments.

To validate the efficiency of distributed computation with PASS-LR2, we compared running times on 6M examples with dimensionality reduced to 1,000 when using 1–22 cores. As shown in Fig. 3b, the speed-up is close to optimal: $K$ cores produces a speedup of about $K/2$ (baseline 3 minutes using 1 core). We used Ray to implement the distributed version of PASS-LR2 [28].[5]

## 6   Discussion

We have presented PASS-GLM, a novel framework for scalable parameter estimation and Bayesian inference in generalized linear models. Our theoretical results provide guarantees on the quality of point estimates as well as approximate posteriors derived from PASS-GLM. We validated our approach empirically with logistic regression and a quadratic approximation. We showed competitive performance on a variety of real-world data, scaling to 40 million examples with 20,000 covariates, and trivial distributed computation with no compounding of approximation error.

There a number of important directions for future work. The first is to use randomization methods along the lines of random projections and random feature mappings [23, 33] to scale to larger $M$ and $d$. We conjecture that the use of randomization will allow experimentation with other GLMs for which quadratic approximations are insufficient.

## Acknowledgments

JHH and TB are supported in part by ONR grant N00014-17-1-2072, ONR MURI grant N00014-11-1-0688, and a Google Faculty Research Award. RPA is supported by NSF IIS-1421780 and the Alfred P. Sloan Foundation.

## Footnotes

[1]Our presentation is slightly different from the standard textbook account because we have implicitly absorbed the base measure and log-partition function into $\mathbf{t}$ and $\boldsymbol{\eta}$.

[2]Recall that a twice-differentiable function $f : \mathbb{R}^d \to \mathbb{R}$ is $\varrho$-strongly convex at $\boldsymbol{\theta}$ if the minimum eigenvalue of the Hessian of $f$ evaluated at $\boldsymbol{\theta}$ is at least $\varrho > 0$.

[3]The Lipschitz constant of function $f : \mathbb{R}^d \to \mathbb{R}$ is $\|f\|_L := \sup_{\mathbf{v}, \mathbf{w} \in \mathbb{R}^d} \frac{\|\phi(\mathbf{v}) - \phi(\mathbf{w})\|_2}{\|\mathbf{v} - \mathbf{w}\|_2}$.

[4]Code is available at `https://bitbucket.org/jhhuggins/pass-glm`.

[5]`https://github.com/ray-project/ray`

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
