[Supplementary Material]

# Supplementary Material for
# *PASS-GLM: polynomial approximate sufficient statistics for scalable Bayesian GLM inference*

**Jonathan H. Huggins**
CSAIL, MIT
jhuggins@mit.edu

**Ryan P. Adams**
Google Brain and Princeton
rpa@princeton.edu

**Tamara Broderick**
CSAIL, MIT
tbroderick@csail.mit.edu

## A   General Derivation of PASS-GLM

We can generalize the setup described in Section 3 to cover a wide range of GLMs by assuming the log-likelihood is of the form

$$\log p(y \mid \mathbf{x}, \boldsymbol{\theta}) = \sum_{k=1}^{K} y^{\alpha_k} \phi_{(k)}(y^{\beta_k} \mathbf{x} \cdot \boldsymbol{\theta} - a_k y),$$

where typically $\alpha_k, \beta_k, a_k \in \{0, 1\}$. We consider the $K = 1$ case and drop the $k$ subscripts since the extension to $K > 1$ is trivial and serves only to introduce extra notational clutter. Letting $\phi_M(s) = \sum_{m=0}^{M} b_m^{(M)} s^m$ be the order $M$ polynomial approximation to $\phi(s) = \phi_{(1)}(s)$, we have that

$$\begin{aligned}
\log p(y \mid \mathbf{x}, \boldsymbol{\theta}) &\approx y^{\alpha} \phi_M(y^{\beta} \mathbf{x} \cdot \boldsymbol{\theta} - ay) \\
&= y^{\alpha} \sum_{m=0}^{M} b_m^{(M)} (y^{\beta} \mathbf{x} \cdot \boldsymbol{\theta} - ay)^m \\
&= y^{\alpha} \sum_{m=0}^{M} b_m^{(M)} \sum_{i=0}^{m} \binom{m}{i} (y^{\beta} \mathbf{x} \cdot \boldsymbol{\theta})^i (ay)^{m-i} \\
&= \sum_{i=0}^{M} (y^{\beta} \mathbf{x} \cdot \boldsymbol{\theta})^i y^{\alpha} \sum_{m=i}^{M} b_m^{(M)} \binom{m}{i} (ay)^{m-i} \\
&= \sum_{i=0}^{M} \sum_{\substack{\mathbf{k} \in \mathbb{N}^d \\ \sum_j k_j = i}} a'(\mathbf{k}, i, M, y) (y^{\beta} \mathbf{x})^{\mathbf{k}} \boldsymbol{\theta}^{\mathbf{k}},
\end{aligned}$$

where $a'(\mathbf{k}, \bar{k}, M, y) := y^{\alpha} \binom{\bar{k}}{\mathbf{k}} \sum_{m=i}^{M} b_m^{(M)} \binom{m}{\bar{k}} (ay)^{m-\bar{k}}$. Thus, we have an exponential family model with

$$\mathbf{t}(\mathbf{x}, y) = \left( a'\left( \mathbf{k}, \sum_j k_j, M, y \right) \mathbf{x}^{\mathbf{k}} \right)_{\mathbf{k}} \qquad \text{and} \qquad \boldsymbol{\eta}(\boldsymbol{\theta}) = (\boldsymbol{\theta}^{\mathbf{k}})_{\mathbf{k}},$$

where $\mathbf{k}$ is taken over all $\mathbf{k} \in \mathbb{N}^d$ such that $\sum_j k_j \leq M$.

The following examples show how a variety of GLM models fit into our framework. Throughout, let $s = \mathbf{x}_n \cdot \boldsymbol{\theta}$.

**Example A.1** (Robust regression)**.** For robust regression, $\mathcal{Y} = \mathbb{R}$ and the log-likelihood is in the form $\phi(s - y)$, where $\phi$ is a choice of "distance" function. For example, we could use either the

Laplace likelihood

$$\phi_{\mathsf{Laplace}}(s - y) := -\frac{|s - y|}{b},$$

the Cauchy likelihood

$$\phi_{\mathsf{Cauchy}}(s - y) := -\ln\left(1 + \frac{(s - y)^2}{b^2}\right),$$

the negative Huber loss

$$\phi_{\mathsf{Huber}}(s - y) := \begin{cases} -\frac{1}{2}(s - y)^2 & |s - y| \leq b \\ -b|s - y| + \frac{1}{2}b^2 & \text{otherwise,} \end{cases}$$

or the negative smoothed Huber loss

$$\phi_{\mathsf{SHuber}}(s - y) := -b^2\left(\sqrt{1 + \frac{(s - y)^2}{b^2}} - 1\right),$$

where in each case $b$ serves as a scale parameter.

**Example A.2** (Poisson regression). For Poisson regression, $\mathcal{Y} = \mathbb{N}$, and the log-likelihood is $ys - e^s$, so $\phi_{(1)}(s) = s$, $\phi_{(2)}(s) = -e^s$, $\alpha_1 = 1$, and $\beta_1 = a_1 = \alpha_2 = \beta_2 = a_2 = 0$.

**Example A.3** (Gamma regression). For gamma regression, $\mathcal{Y} = \mathbb{R}_+$, and the log-likelihood is $-\nu s - \nu y e^{-s} + c(y, \nu)$ if using the log link, where $\nu$ is a scale parameter. We can ignore the $c(y, \nu)$ term since it does not depend on $\boldsymbol{\theta}$. Thus, $\phi_{(1)}(s) = -\nu s$, $\phi_{(2)}(s) = -\nu e^{-s}$, $\alpha_2 = 1$, and $\beta_1 = a_1 = \alpha_1 = \beta_2 = a_2 = 0$.

**Example A.4** (Probit regression). For probit regression, $\mathcal{Y} = \{0, 1\}$, and the log-likelihood is

$$\begin{cases} \ln(1 - \Phi(s)) & z = 0 \\ \ln(\Phi(s)) & z = 1 \end{cases},$$

where $\Phi$ denotes the standard normal CDF. Thus, $\phi_{(1)}(s) = \ln(1 - \Phi(s))$, $\phi_{(2)}(s) = \ln(\Phi(s)) - \ln(1 - \Phi(s))$, $\alpha_2 = 1$, and $\beta_1 = a_1 = \alpha_1 = \beta_2 = a_2 = 0$.

# B  Chebyshev Approximation Results

We begin by summarizing some standard results on the approximation accuracy of Chebyshev polynomials. Let $\phi : [-1, 1] \to \mathbb{R}$ be a continuous function, and let $\phi_M$ be the $M$-th order Chebyshev approximation to $\phi$. Let $\|f\|_\infty := \sup_s |f(s)|$ be the $L^\infty$ norm of a function $f$; let $\mathbb{C}$ denote the set of complex numbers; and let $|z|$ be the absolute value of $z \in \mathbb{C}$.

**Theorem B.1** (Mason and Handscomb [5, Theorem 5.14]). *If $\phi$ has $k + 1$ continuous derivatives, then $\|\phi - \phi_M\|_\infty = O(M^{-k})$.*

**Theorem B.2** (Mason and Handscomb [5, Theorem 5.16]). *If $\phi$ can be extended to an analytic function on $E_r := \{z \in \mathbb{C} : |z + \sqrt{z^2 - 1}| = r\}$ for $r > 1$ and $C := \sup_{z \in E_r} |\phi(z)|$, then*

$$\|\phi - \phi_M\|_\infty \leq \frac{C}{r - 1}r^{-M}.$$

Chebyshev polynomials also provide a uniformly good approximation of the derivative of the function they are used to approximate.

**Theorem B.3.** *If $\phi$ can be extended to an analytic function on $E_r$ for $r > 1$ and $C := \sup_{z \in E_r} |\phi(z)|$, then*

$$\|\phi' - \phi_M'\|_\infty \leq Cr^{-M}\frac{r + 1}{(r - 1)^4}\left[M^2 r(r + 1) + M(2r^2 + r + 1) + r(r + 1)\right] =: B(C, r, M)$$

*Proof.* The proof follows the same structure as that for Theorem 5.16 in Mason and Handscomb [5]. For Chebyshev polynomials, $\varsigma(\mathrm{d}s) = \frac{2}{\pi}(1 - s^2)^{-1/2}\mathrm{d}s$. Note that $\phi(s) = \sum_{m=0}^{\infty}(\int \phi\psi_m \mathrm{d}\varsigma)\psi_m(s)$

and hence $\phi'(s) = \sum_{m=0}^{\infty}(\int \phi \psi_m \mathrm{d}\varsigma)\psi'_m(s)$. Since $\psi'_m = mU_{m-1}$, where $\{U_m\}_{m\geq0}$ are the Chebyshev polynomials of the second kind,

$$\phi'(s) - \phi'_M(s) = \sum_{m=M+1}^{\infty} \frac{2m}{\pi} \int_{-1}^{1} (1-v^2)^{-1/2}\phi(v)\psi_m(v)U_{m-1}(s)\mathrm{d}v.$$

Define the conformal mappings $s = \frac{1}{2}(\xi + \xi^{-1})$ and $v = \frac{1}{2}(\zeta + \zeta^{-1})$, and $\phi(v) =: \tilde{\phi}(\zeta) = \tilde{\phi}(\zeta^{-1})$. By assumption, $|\tilde{\phi}(\zeta)| \leq C$. Let $\mathcal{C}_1$ denote the complex unit circle and for $r \in \mathbb{R}_+$, let $\mathcal{C}_r := r\mathcal{C}_1$. Using the conformal mappings, we have

$$\phi'(s) - \phi'_M(s)$$

$$= \sum_{m=M+1}^{\infty} \frac{m}{4\mathrm{i}\pi} \oint_{\mathcal{C}_1} \tilde{\phi}(\zeta)(\zeta^m + \zeta^{-m})\frac{\xi^m - \xi^{-m}}{\xi - \xi^{-1}}\frac{\mathrm{d}\zeta}{\zeta}$$

$$= \sum_{m=M+1}^{\infty} \frac{m}{2\mathrm{i}\pi} \oint_{\mathcal{C}_r} \tilde{\phi}(\zeta)\zeta^{-m}\frac{\xi^m - \xi^{-m}}{\xi - \xi^{-1}}\frac{\mathrm{d}\zeta}{\zeta}$$

$$= \frac{1}{2\mathrm{i}\pi} \oint_{\mathcal{C}_r} \frac{\tilde{\phi}(\zeta)}{\xi - \xi^{-1}} \left( \frac{\xi^{M+1}\zeta^{-M-1}(1 + M + \xi\zeta^{-1})}{(\xi\zeta^{-1} - 1)^2} - \frac{\xi^{-M-1}\zeta^{-M-1}(1 + M + \xi^{-1}\zeta^{-1})}{(\zeta^{-1}\xi^{-1} - 1)^2} \right) \frac{\mathrm{d}\zeta}{\zeta}$$

$$\leq \frac{C}{2\mathrm{i}\pi} \oint_{\mathcal{C}_r} \frac{\xi\zeta^{-M-1}\xi^{-M-1}}{\xi^2 - 1} \left( \frac{\xi^{2M+2}(1 + M + \xi\zeta^{-1})}{(\xi\zeta^{-1} - 1)^2} - \frac{(1 + M + \xi^{-1}\zeta^{-1})}{(\zeta^{-1}\xi^{-1} - 1)^2} \right) \frac{\mathrm{d}\zeta}{\zeta}.$$

Letting $\eta := \xi^2$ and $\psi := \xi^{-1}\zeta^{-1}$, the absolute value of the integrand is

$$\frac{|\psi|^{M+1}}{|\eta - 1|} \left| \frac{\eta^{M+1}(1 + M - \eta\psi)}{(\eta\psi - 1)^2} - \frac{1 + M - \psi}{(\psi - 1)^2} \right|$$

$$= r^{-M-1}\frac{|\eta\psi - 1|^{-2}|\psi - 1|^{-2}}{|\eta - 1|} \left| \eta^{M+1}(1 + M - \eta\psi)(\psi - 1)^2 - (1 + M - \psi)(\eta\psi - 1)^2 \right|$$

$$\leq r^{-M-1}\frac{(r^{-1} - 1)^{-4}}{|\eta - 1|} \Big[ |\psi||\eta^{M+2} - 1| + (M+1)|\eta^{M+1} - 1| + 2|\psi|^2|\eta^{M+1} - 1|$$

$$+ 2(M+1)|\psi||\eta^M - 1| + |\psi|^3|\eta^M - 1| + (M+1)|\phi|^2|\eta^{M-1} - 1| \Big]$$

$$\leq \frac{r^{-M+3}}{(r-1)^4} \left[ \frac{M+2}{r} + (M+1)^2 + \frac{2(M+1)}{r^2} + \frac{2M(M+1)}{r} + \frac{M}{r^3} + \frac{M^2 - 1}{r^2} \right]$$

$$= r^{-M}\frac{r+1}{(r-1)^4} \left[ M^2 r(r+1) + M(2r^2 + r + 1) + r(r+1) \right].$$

The final inequality follows from the fact that for $k \in \mathbb{N}$,

$$|\eta^k - 1|/|\eta - 1| = |\sin(k\arg(\eta))/\sin(\arg(\eta))| \leq k.$$

The result now follows. $\qquad \square$

Since $\phi_{\mathrm{logit}}$ is smooth, we can apply Theorems B.2 and B.3 to obtain exponential convergence rates of the (derivative of the) Chebyshev approximation. The same is true in the Poisson and smoothed Huber regression cases.

**Corollary B.4.** *Fix $R > 0$. If $\phi(s) = \log(1 + e^{-Rs})$, $s \in [-1, 1]$, then for any $r \in (1, \pi/R + \sqrt{\pi^2/R^2 + 1}\,)$,*

$$\|\phi - \phi_M\|_\infty \leq \frac{C(r, R)}{(r-1)r^M} \qquad and \qquad \|\phi' - \phi'_M\|_\infty \leq B(C(r, R), r, M),$$

*where $C(r, R) := \left|\log\left(1 + e^{-\frac{1}{2}R(r-r^{-1})i}\right)\right|$.*

*Proof.* The function $e^{-Rs}$ is entire while $\log$ is analytic except at $0$. Thus, we must determine the minimum value of $r$ such that there exists $z \in E_r$ such that $1 + e^{-Rz} = 0$. Taking $z = a + bi$, it

must hold that $b \in \{k\pi/R : k \in \mathbb{Z}\}$ since otherwise $e^{-Rz}$ would contain an imaginary component. If $b = 2k\pi/R$ then $e^{-Rz} = e^{-Ra} > 0$, so this cannot be a solution to $1 + e^{-Rz} = 0$. However, taking $b = (2k+1)\pi/R$ yields $1 - e^{-Ra} = 0 \implies a = 0$. Hence, $z = (2k+1)\pi\mathrm{i}/R$ and thus

$$
\begin{aligned}
|z + \sqrt{z^2 - 1}| &= |\pi\mathrm{i}/R + \sqrt{-(2k+1)^2\pi^2/R^2 - 1}| \\
&= |(\pi/R + \sqrt{(2k+1)^2\pi^2/R^2 + 1})\mathrm{i}| \\
&= \pi/R + \sqrt{(2k+1)^2\pi^2/R^2 + 1} \\
&\geq \pi/R + \sqrt{\pi^2/R^2 + 1}.
\end{aligned}
$$

Thus we must choose $r < \pi/R + \sqrt{\pi^2/R^2 + 1}$. For any such $r$, $|\phi(z)|$ is maximized along $E_r$ when $z = b\mathrm{i}$, which implies $b = \frac{1}{2}(r - r^{-1})$ and hence $C = C(r, R)$. The two inequalities now follow from, respectively, Theorems B.2 and B.3. $\qquad\square$

**Corollary B.5.** *Fix $R > 0$. If $\phi(s) = e^{Rs}$, $s \in [-1, 1]$, then for any $r > 1$,*

$$
\|\phi - \phi_M\|_\infty \leq \frac{e^{\frac{1}{2}R(r+r^{-1})}}{(r-1)r^M}
$$

$$
\|\phi' - \phi'_M\|_\infty \leq B(e^{\frac{1}{2}R(r+r^{-1})}, r, M).
$$

*Proof.* The proof is similar to that for Corollary B.4. The differences are as follows. The function $e^{-Rs}$ is entire, so we may choose any $r > 1$. For any such $r$, $|\phi(z)|$ is maximized along $E_r$ when $z$ is real, which implies $z = \frac{1}{2}(r + r^{-1})$ and hence $C = e^{\frac{1}{2}R(r+r^{-1})}$. $\qquad\square$

**Corollary B.6.** *Fix $R > 0$. If $\phi(s) = b^2\left(\sqrt{1 + \frac{R^2 s^2}{b^2}} - 1\right)$, $s \in [-1, 1]$, then for any $r \in (1, b/R + \sqrt{b^2/R^2 + 1})$,*

$$
\|\phi - \phi_M\|_\infty \leq \frac{b^2\sqrt{1 + \{(r^2+1)/(2rb)\}^2} - b^2}{r - 1} r^{-M}
$$

$$
\|\phi' - \phi'_M\|_\infty \leq B\left(b^2\sqrt{1 + \{(r^2+1)/(2rb)\}^2} - b^2, r, M\right).
$$

*Proof.* The proof is similar to that for Corollary B.4. The differences are as follows. The square root function is analytic except at zero, so we must determine the minimum value of $r$ such that there exists $z \in E_r$ such that $1 + R^2 z^2/b^2 = 0$. Solving, we find that $z = \mathrm{i}b/R$. Thus, we have

$$
|z + \sqrt{z^2 - 1}| = b/R + \sqrt{b^2/R^2 + 1}
$$

and so must choose $1 < r < b/R + \sqrt{b^2/R^2 + 1}$. For any such $r$, $|\phi(z)|$ is maximized along $E_r$ when $z$ is real, which implies $z = \frac{r^2+1}{2r}$ and hence $C = b^2\left(\sqrt{1 + \left(\frac{r^2+1}{2rb}\right)^2} - 1\right)$. $\qquad\square$

# C  Approximation Theorems and Proofs

**Theorem C.1.** *Let $\mathbb{B}_r(\boldsymbol{\theta}^*) := \{\boldsymbol{\theta} \in \Theta \mid \|\boldsymbol{\theta} - \boldsymbol{\theta}^*\|_2 \leq r\}$. Assume there exist parameters $\varepsilon_N$ and $\varrho_N$ such that for all $\boldsymbol{\theta} \in \mathbb{B}_{r_N}(\boldsymbol{\theta}_{MAP})$, where $r_N^2 := 4\varepsilon_N/\varrho_N$,*

*(A) $|\mathcal{L}_\mathcal{D}(\boldsymbol{\theta}) - \tilde{\mathcal{L}}_\mathcal{D}(\boldsymbol{\theta})| \leq \varepsilon_N$ and*  (B) *$-\log \pi_\mathcal{D}$ is $\varrho_N$-strongly convex.[1]*

*Furthermore, assume that for all $\boldsymbol{\theta} \in \Theta$,*

*(C) $\log \pi_\mathcal{D}$ is strictly quasi-concave[2] and*  (D) *$\tilde{\mathcal{L}}_\mathcal{D}(\boldsymbol{\theta}) \leq \mathcal{L}_\mathcal{D}(\boldsymbol{\theta}) + \varepsilon_N$.*

*Then* $\|\boldsymbol{\theta}_{MAP} - \tilde{\boldsymbol{\theta}}_{MAP}\|_2^2 \le \frac{4\varepsilon_N}{\varrho_N}$.

*Remark* (Assumptions). The error in the MAP estimate naturally depends on the error of the approximate log-likelihood (Assumption (A)) as well as the flatness of the posterior (Assumption (B)). In the latter case, if $\log \pi_{\mathcal{D}}$ is very flat, then even a small error from using $\tilde{\mathcal{L}}_{\mathcal{D}}$ in place of $\mathcal{L}_{\mathcal{D}}$ could lead to a large error in the approximate MAP solution. However, the stronger assumptions, (A) and (B), need hold only near the MAP solution.

*Remark* (Strict quasi-concavity). Requiring that $\log \pi_{\mathcal{D}}$ be only strictly quasi-concave (rather than strongly log-concave everywhere) substantially increases the applicability of the result. For instance, it allows heavy-tailed priors (e.g., Cauchy) as well as sparsity-inducing priors (e.g., Laplace/$L_1$ regularization).

*Proof of Theorem C.1.* An equivalent condition for $f$ to be strictly quasi-convex is that if $f(\mathbf{v}) > f(\mathbf{w})$ then $\langle \nabla f(\mathbf{w}), \mathbf{v} - \mathbf{w} \rangle > 0$ [6, Theorem 21.14]. We obtain the result by considering some $\boldsymbol{\theta}$ such that $\boldsymbol{\theta} \notin \mathbb{B}_{r_N}(\boldsymbol{\theta}_{\text{MAP}})$. Since $\varpi := \log \pi_{\mathcal{D}}$ is strictly quasi-concave (by Assumption (C)), if it has a global maximum it is unique (if it had two global maxima, this would immediately yield a contradiction). By hypothesis $\boldsymbol{\theta}_{\text{MAP}}$ is such a global maximum. Thus, $\varpi(\boldsymbol{\theta}_{\text{MAP}}) > \varpi(\boldsymbol{\theta})$, which implies

$$\langle \nabla \varpi(\boldsymbol{\theta}), \boldsymbol{\theta}_{\text{MAP}} - \boldsymbol{\theta} \rangle > 0. \tag{C.1}$$

Now, fix $\boldsymbol{\theta}'$ such that $\boldsymbol{\theta}' \notin \mathbb{B}_{r_N}(\boldsymbol{\theta}_{\text{MAP}})$. Let $r'_N := \|\boldsymbol{\theta}' - \boldsymbol{\theta}_{\text{MAP}}\|_2 > r_N$ and $\boldsymbol{\theta}'' := \frac{r_N}{r'_N}\boldsymbol{\theta}' + \frac{r'_N - r_N}{r'_N}\boldsymbol{\theta}_{\text{MAP}}$, the projection of $\boldsymbol{\theta}'$ onto $\mathbb{B}_{r_N}(\boldsymbol{\theta}_{\text{MAP}})$. Applying the fundamental theorem of calculus for line integrals on the linear path $\gamma[\boldsymbol{\theta}', \boldsymbol{\theta}'']$ from $\boldsymbol{\theta}'$ to $\boldsymbol{\theta}''$, parameterized as $\boldsymbol{\theta}(t) = t\boldsymbol{\theta}'' + (1-t)\boldsymbol{\theta}'$, we have

$$
\begin{aligned}
\mathcal{L}_{\mathcal{D}}(\boldsymbol{\theta}'') - \mathcal{L}_{\mathcal{D}}(\boldsymbol{\theta}') &= \int_{\gamma[\boldsymbol{\theta}', \boldsymbol{\theta}'']} \nabla \varpi(\boldsymbol{\theta}) \cdot \mathrm{d}\boldsymbol{\theta} \\
&= \int_0^1 \nabla \varpi(\boldsymbol{\theta}(t)) \cdot (\boldsymbol{\theta}'' - \boldsymbol{\theta}') \, \mathrm{d}t \\
&= \frac{r'_N - r_N}{r'_N} \int_0^1 \nabla \varpi(\boldsymbol{\theta}(t)) \cdot (\boldsymbol{\theta}_{\text{MAP}} - \boldsymbol{\theta}') \, \mathrm{d}t \\
&= \frac{r'_N - r_N}{r'_N} \int_0^1 C(t) \nabla \varpi(\boldsymbol{\theta}(t)) \cdot (\boldsymbol{\theta}_{\text{MAP}} - \boldsymbol{\theta}(t)) \mathrm{d}t \\
&> 0,
\end{aligned}
$$

where $C(t) := \frac{r'_N}{r'_N - tr'_N + tr_N}$ and the inequality follows from Eq. (C.1). Hence,

$$\varpi(\boldsymbol{\theta}') < \varpi(\boldsymbol{\theta}'') \tag{C.2}$$

and

$$
\begin{aligned}
\log \pi_0(\boldsymbol{\theta}') + \tilde{\mathcal{L}}_{\mathcal{D}}(\boldsymbol{\theta}') &\le \log \pi_0(\boldsymbol{\theta}') + \mathcal{L}_{\mathcal{D}}(\boldsymbol{\theta}') + \varepsilon_N && \text{by Assumption (D)} \\
&< \log \pi_0(\boldsymbol{\theta}'') + \mathcal{L}_{\mathcal{D}}(\boldsymbol{\theta}'') + \varepsilon_N && \text{by Eq. (C.2)} \\
&\le \log \pi_0(\boldsymbol{\theta}_{\text{MAP}}) + \mathcal{L}_{\mathcal{D}}(\boldsymbol{\theta}_{\text{MAP}}) + \varepsilon_N - \frac{\varrho_N r_N^2}{2} && \text{by Assumption (B)} \\
&= \log \pi_0(\boldsymbol{\theta}_{\text{MAP}}) + \mathcal{L}_{\mathcal{D}}(\boldsymbol{\theta}_{\text{MAP}}) - \varepsilon_N && \text{by definition of } r_n \\
&\le \log \pi_0(\boldsymbol{\theta}_{\text{MAP}}) + \tilde{\mathcal{L}}_{\mathcal{D}}(\boldsymbol{\theta}_{\text{MAP}}) && \text{by Assumption (A)}.
\end{aligned}
$$

So $\boldsymbol{\theta}'$ is not a global optimum of $\log \tilde{\pi}_{\mathcal{D}}$ and hence $\tilde{\boldsymbol{\theta}}_{\text{MAP}} \in \mathbb{B}_{R_N}(\boldsymbol{\theta}_{\text{MAP}})$. $\qquad \square$

We present a generalization of Corollary 4.1. Let $\|\mathbf{T}\|_{op} := \sup_{\substack{\mathbf{v} \in \mathbb{R}^d \\ \|\mathbf{v}\|_2 = 1}} \|\mathbf{T}[\mathbf{v}]\|_{op}$ denote the operator norm of the tensor $\mathbf{T}$ (with $\|\mathbf{T}\|_{op} = \|\mathbf{T}\|_2$ if $\mathbf{T}$ is a matrix). Recall the Lipschitz operator bound property

$$\|\nabla h(x)\|_{op} = \sup_{y \ne x} \frac{\|h(x) - h(y)\|_{op}}{\|x - y\|_2}, \tag{C.3}$$

which holds for any sufficiently smooth $h : \mathbb{R}^d \to (\mathbb{R}^d)^{\otimes k}$. Recall also that for compatible operators $T$ and $T'$, $\|TT'\|_{op} \leq \|T\|_{op}\|T'\|_{op}$.

**Corollary C.2.** *Assume the tensor defined by $T_{ijk} := \sum_{n=1}^{N} x_{ni}x_{nj}x_{nk}$ satisfies $\|\mathbf{T}\|_{op} \leq LN/d^2$. For the logistic regression model, assume that $\|\nabla^2 \mathcal{L}_{\mathcal{D}}(\boldsymbol{\theta}_{MAP})^{-1}\|_2 \leq cd/N$ and that $\|\mathbf{x}_n\|_2 \leq 1$ for all $n = 1, \ldots, N$. Let $\phi_M$ be the order $M$ Chebyshev approximation to $\phi_{\text{logit}}$ on $[-R, R]$ such that Eq. (2) holds. Let $\tilde{\pi}_{\mathcal{D}}(\boldsymbol{\theta})$ denote the posterior approximation obtained by using $\phi_M$ with a strictly quasi-log concave prior. Let*

$$\varepsilon := \min_{r \in (1, \pi/R + \sqrt{\pi^2/R^2 + 1})} \left| \log\left(1 + e^{-\frac{1}{2}R(r - r^{-1})i}\right) \right| (r - 1)^{-1} r^{-M}$$

*and $\alpha^* := 1 + b - \sqrt{(b+1)^2 - 1}$, where $b := \frac{\varepsilon L^2 c^3}{54d}$. If $R - \|\boldsymbol{\theta}_{MAP}\|_2 \geq 2\sqrt{\frac{cd\varepsilon}{\alpha^*}}$, then*

$$\|\boldsymbol{\theta}_{MAP} - \tilde{\boldsymbol{\theta}}_{MAP}\|_2^2 \leq \frac{4cd\varepsilon}{\alpha^*} \leq \frac{4}{27}c^4 L^2 \varepsilon^2 + 8cd\varepsilon$$

*and Corollary 4.1 follows from the upper bound $\|\mathbf{T}\|_{op} \leq N$ (using the assumption that $\|\mathbf{x}_n\|_2 \leq 1$).*

*Proof.* By Corollary B.4, for all $s \in [-R, R]$, $|\phi_{\text{logit}}(s) - \phi_M(s)| \leq \varepsilon N$. It is easy to verify that $\max_{s \in \mathbb{R}} |\phi'''_{\text{logit}}(s)| = \frac{1}{6\sqrt{3}}$ and therefore $\|\nabla^3 \mathcal{L}_{\mathcal{D}}(\boldsymbol{\theta})\|_{op} \leq \frac{1}{6\sqrt{3}}\|\mathbf{T}\|_{op} \leq \frac{LN}{6\sqrt{3}d^2}$. Since by hypothesis $\|(\nabla^2 \mathcal{L}_{\mathcal{D}}(\boldsymbol{\theta}_{\text{MAP}}))^{-1}\|_2 \leq cd/N$, $\mathcal{L}_{\mathcal{D}}(\boldsymbol{\theta}_{\text{MAP}})$ is $N/(cd)$-strongly concave. We can write $\nabla(\nabla^2 \mathcal{L}_{\mathcal{D}})^{-1} = -(\nabla^2 \mathcal{L}_{\mathcal{D}})^{-1}\nabla^3 \mathcal{L}_{\mathcal{D}}(\nabla^2 \mathcal{L}_{\mathcal{D}})^{-1}$ if we treat the first $(\nabla^2 \mathcal{L}_{\mathcal{D}})^{-1}$ as a matrix to matrix operator, $\nabla^3 \mathcal{L}_{\mathcal{D}}$ as a vector to matrix operator, and the second $(\nabla^2 \mathcal{L}_{\mathcal{D}})^{-1}$ as a vector to vector operator. Thus

$$\|\nabla(\nabla^2 \mathcal{L}_{\mathcal{D}})^{-1}(\boldsymbol{\theta})\|_{op} \leq \|(\nabla^2 \mathcal{L}_{\mathcal{D}})^{-1}(\boldsymbol{\theta})\|_{op}^2 \|\nabla^3 \mathcal{L}_{\mathcal{D}}(\boldsymbol{\theta})\|_{op} \leq \frac{c^2 d^2}{N^2} \frac{LN}{6\sqrt{3}d^2} = \frac{c^2 L}{6\sqrt{3}N}.$$

Using the triangle inequality and Eq. (C.3), we have

$$
\begin{aligned}
\|(\nabla^2 \mathcal{L}_{\mathcal{D}})^{-1}(\boldsymbol{\theta})\|_{op} &\leq \|(\nabla^2 \mathcal{L}_{\mathcal{D}})^{-1}(\boldsymbol{\theta}_{\text{MAP}})\|_{op} + \|(\nabla^2 \mathcal{L}_{\mathcal{D}})^{-1}(\boldsymbol{\theta}) - (\nabla^2 \mathcal{L}_{\mathcal{D}})^{-1}(\boldsymbol{\theta}_{\text{MAP}})\|_{op} \\
&\leq \|(\nabla^2 \mathcal{L}_{\mathcal{D}})^{-1}(\boldsymbol{\theta}_{\text{MAP}})\|_{op} + \|\nabla(\nabla^2 \mathcal{L}_{\mathcal{D}})^{-1}(\boldsymbol{\theta})\|_{op}\|\boldsymbol{\theta} - \boldsymbol{\theta}_{\text{MAP}}\|_2 \\
&\leq \frac{cd}{N} + \frac{c^2 L}{6\sqrt{3}N}\|\boldsymbol{\theta} - \boldsymbol{\theta}_{\text{MAP}}\|_2,
\end{aligned}
$$

so $\mathcal{L}_{\mathcal{D}}(\boldsymbol{\theta})$ is $\alpha N/(cd)$-strongly concave for all $\boldsymbol{\theta} \in \mathbb{B}_\Delta(\boldsymbol{\theta}_{\text{MAP}})$ if

$$\frac{cd}{N} + \frac{c^2 L\Delta}{6\sqrt{3}N} \leq \frac{cd}{N\alpha} \qquad \Longleftrightarrow \qquad \Delta^2 \leq \frac{108d^2(1 - \alpha)^2}{L^2 c^2 \alpha^2}.$$

To apply Theorem C.1, we require that $\Delta^2 \geq 4\varepsilon cd/\alpha$. Combining the two inequalities, we have

$$\frac{4\varepsilon cd}{\alpha} \leq \frac{108d^2(1 - \alpha)^2}{L^2 c^2 \alpha^2} \qquad \Longleftrightarrow \qquad \frac{\varepsilon c^3 L^2}{27d}\alpha \leq (1 - \alpha)^2 \qquad \Longleftrightarrow \qquad 0 \leq \alpha^2 - (2 + b)\alpha + 1.$$

Solving the quadratic implies that the maximal viable $\alpha$ value is $\alpha^* = 1 + b - \sqrt{(b+1)^2 - 1} \geq \frac{1}{2(b+1)}$.

Requiring $R - \|\boldsymbol{\theta}_{\text{MAP}}\|_2 \geq 2\sqrt{\frac{cd\varepsilon}{\alpha^*}}$ together with the hypothesis that $\|\mathbf{x}_n\| \leq 1$ ensures that we are considering only inner products $\mathbf{x}_n \cdot \boldsymbol{\theta} \in [-R, R]$. Since Eq. (2) holds by hypothesis, Assumption (D) holds. The result now follows from Theorem C.1. $\square$

*Proof sketch of Corollary 4.3.* The proof is similar in spirit to Corollary C.2. The key differences are that we apply Corollary B.6 and use the condition that a constant fraction of the data satisfies $|\mathbf{x}_n \cdot \boldsymbol{\theta}_{\text{MAP}} - y_n| \leq b/2$ to guarantee $\Theta(N)$-strong log-convexity of $-\log \pi_{\mathcal{D}}$ near the MAP. $\square$

Recall that a centered random variable $X$ is said to be $\sigma^2$-*subgaussian* [1, Section 2.3] if for all $s \in \mathbb{R}$,

$$\mathbb{E}[e^{sX}] \leq e^{s^2 \sigma^2/2}.$$

**Theorem C.3.** *Assume that*

*(E)* $-\log \tilde{\pi}_{\mathcal{D}}(\boldsymbol{\theta})$ *is $\tilde{\varrho}$-strongly convex,*

*(F) for all $n = 1, \ldots, N$, $\|\mathbf{x}_n\|_2 \leq 1$,*

*(G) there exist constants $a_n, b, R, \alpha \in \mathbb{R}_+$ such that*

$$\|\nabla_{\boldsymbol{\theta}}\phi(\langle y_n\mathbf{x}_n, \boldsymbol{\theta}\rangle) - \nabla_{\boldsymbol{\theta}}\phi_M(\langle y_n\mathbf{x}_n, \boldsymbol{\theta}\rangle)\|_2 \leq a_n + b\max(0, |\langle y_n\mathbf{x}_n, \boldsymbol{\theta}\rangle| - R), \text{ and}$$

*(H)* $-\log \pi_{\mathcal{D}}(\boldsymbol{\theta})$ *is $\varrho$-strongly convex with mean $\bar{\boldsymbol{\theta}}$.*

*Let $\sigma_1, \sigma_2$ be the subgaussianity constants of, respectively, the random variables $\langle y_n\mathbf{x}_n, \bar{\boldsymbol{\theta}}\rangle - \delta_1$ and $\|y_n\mathbf{x}_n\|_2^2 - \delta_2$, where the randomness is over $n \sim \mathsf{Unif}\{1, \ldots, N\}$. Let $\delta_1 := \mathbb{E}[\langle y_n\mathbf{x}_n, \bar{\boldsymbol{\theta}}\rangle]$, $\delta_2 := \mathbb{E}[\|y_n\mathbf{x}_n\|_2^2]$, and $\bar{a} := \sum_{n=1}^N a_n$. Then there exists an explicit constant $\varepsilon$ (equal to zero if $b = 0$ and depending on $R$, $\varrho$, $\sigma_1$, $\sigma_2$, $\delta_1$, and $\delta_2$ otherwise) such that*

$$d_{\mathcal{W}}(\pi_{\mathcal{D}}, \tilde{\pi}_{\mathcal{D}}) \leq \tilde{\varrho}^{-1}(\bar{a} + Nb\varepsilon).$$

*Remark* (Value of $\varepsilon$). The definition of the constant $\varepsilon$ is given in the proof of the theorem.

*Remark* (Assumptions). Our posterior approximation result primarily depends on the peakedness of the approximate posterior (Assumption (E)) and the error of the approximate gradients (Assumption (G)). If the gradients are poorly approximated then the error can be large while if the (approximate) posterior is flat then even small likelihood errors could lead to large shifts in expected values of the parameters and hence large Wasserstein error.

*Remark* (Verifying assumptions). In the corollaries we use Theorem B.3 to control the gradient error in the case of Chebyshev polynomial approximations, which allows us to satisfy Assumption (G). Whether Assumption (E) holds will depend on the choices of $M$, $\phi$, and $\pi_0$. For example, if $M = 2$ and $-\log \pi_0$ is convex, then the assumption holds. This assumption could be relaxed to only assume, e.g., a "bounded concavity" condition along with strong convexity in the tails. See Eberle [2], Gorham et al. [3, Section 4], and Huggins and Zou [4, Appendix A] for full details. It is possible that Assumption (H) could also be weakened. The key is to have some control of the tails of $\pi_{\mathcal{D}}$. Both $\langle y_n\mathbf{x}_n, \bar{\boldsymbol{\theta}}\rangle$ and $\|y_n\mathbf{x}_n\|_2^2$ are subgaussian since $y_n\mathbf{x}_n$ is bounded.

*Proof of Theorem C.3.* By Assumption (G), we have that

$$\begin{aligned}
\mathrm{err}(\boldsymbol{\theta}) &:= \|\nabla\log\pi_{\mathcal{D}}(\boldsymbol{\theta}) - \nabla\log\tilde{\pi}_{\mathcal{D}}(\boldsymbol{\theta})\|_2 \\
&\leq \sum_{n=1}^N \|\nabla_{\boldsymbol{\theta}}\phi(\langle y_n\mathbf{x}_n, \boldsymbol{\theta}\rangle) - \nabla_{\boldsymbol{\theta}}\phi_M(\langle y_n\mathbf{x}_n, \boldsymbol{\theta}\rangle)\|_2 \\
&\leq \bar{a} + \sum_{n=1}^N b\max(0, |\langle y_n\mathbf{x}_n, \boldsymbol{\theta}\rangle| - R).
\end{aligned}$$

By Lemma C.4, the random variable $W := \langle y_n\mathbf{x}_n, \boldsymbol{\theta}\rangle - \delta_1$ is $(\lambda, \beta)$-subexponential. Hence for $t \geq 0$,

$$\mathbb{P}(W \geq t) \vee \mathbb{P}(W - \delta \leq -t) \leq \bar{p}(t, \lambda, \beta) := e^{-\left(\frac{t^2}{2\lambda^2} \wedge \frac{t}{2\beta}\right)}.$$

We can now bound $\pi_{\mathcal{D}}(\mathrm{err})$:

$$\begin{aligned}
\pi_{\mathcal{D}}(\mathrm{err}) &\leq aN + \sum_{n=1}^N \mathbb{E}_{\boldsymbol{\theta}\sim\pi_{\mathcal{D}}}[b\max(0, |\langle y_n\mathbf{x}_n, \boldsymbol{\theta}\rangle| - R)]. \\
&= aN + bN\mathbb{E}_{n\sim\mathsf{Unif}\{1,\ldots,N\}}\mathbb{E}_{\boldsymbol{\theta}\sim\pi_{\mathcal{D}}}[\max(0, |\langle y_n\mathbf{x}_n, \boldsymbol{\theta}\rangle| - R)) \\
&= aN + bN\mathbb{E}[\max(0, |W + \delta_1| - R))] \\
&= aN + bN\mathbb{E}[(W + \delta_1 + R)\mathbb{1}(W + \delta_1 \leq -R) + (W + \delta_1 - R)\mathbb{1}(W + \delta_1 \geq R)].
\end{aligned}$$

$$\text{(C.4)}$$

For the second term in the expectation, we have

$$\mathbb{E}[(W + \delta_1 - R)\mathbb{1}(W \geq R - \delta_1)]$$

$$= \int_{R-\delta_1}^{\infty} (w + \delta_1 - R)p(\mathrm{d}w)$$

$$= \int_{R-\delta}^{\infty} \mathbb{P}(W \geq t)\,\mathrm{d}t$$

$$\leq 0 \vee (\delta_1 - R) + \int_{0\vee(R-\delta_1)}^{\infty} \bar{p}(t, \lambda, \beta)\mathrm{d}t =: B(R, \delta_1, \lambda, \beta),$$

By symmetry, the first term in the expectation in Eq. (C.4) is bounded by $B(R, -\delta_1, \lambda, \beta)$, so

$$\pi_{\mathcal{D}}(\mathrm{err}) \leq \bar{a} + Nb(B(R, \delta_1, \lambda, \beta) + B(R, -\delta_1, \lambda, \beta)).$$

Assumption (E) implies that $\tilde{\pi}_{\mathcal{D}}$ satisfies Assumption 2.A of Huggins and Zou [4] with $C = 1$ and $\rho = e^{-\tilde{\varrho}}$. By Theorem 2 of Gorham et al. [3], it is not necessary for the Lipschitz conditions in Assumption 2.A of Huggins and Zou [4] to hold. Furthermore, it can easily be seen that 2.B(3) of Huggins and Zou [4] is not necessary if both $\pi_{\mathcal{D}}$ and $\tilde{\pi}_{\mathcal{D}}$ are strongly convex. The remaining portions of Assumption 2.B of Huggins and Zou [4] are satisfied, however. Thus we can apply Theorem 3.4 from Huggins and Zou [4], which yields

$$d_{\mathcal{W}}(\pi_{\mathcal{D}}, \tilde{\pi}_{\mathcal{D}}) \leq \tilde{\varrho}^{-1}\pi_{\mathcal{D}}(\mathrm{err}) \leq \tilde{\varrho}^{-1}(\bar{a} + Nb\varepsilon),$$

where $\varepsilon := B(R, \delta_1, \lambda, \beta) + B(R, -\delta_1, \lambda, \beta)$. $\qquad \square$

**Lemma C.4.** *Under the conditions of Theorem C.3, the random variable $\langle y_n\mathbf{x}_n, \boldsymbol{\theta}\rangle - \delta_1$ is $(\lambda, \beta)$-subexponential, where $\lambda^2 := 4\left(\frac{1+\delta_2}{\varrho} \vee \sigma_1^2\right)$ and $\beta^2 := \frac{2\sigma_2^2}{\varrho}$.*

*Proof.* Let $z_n = y_n\mathbf{x}_n$. For $|s| \leq 1/\beta$, we have

$$\mathbb{E}[e^{s(\langle z_n, \boldsymbol{\theta}\rangle - \delta_1)}] = \mathbb{E}[\mathbb{E}[e^{s\langle z, \boldsymbol{\theta} - \bar{\boldsymbol{\theta}}\rangle} \mid z_n = z]e^{s(\langle \bar{\boldsymbol{\theta}}, z_n\rangle - \delta_1)}]$$

$$\leq \mathbb{E}[e^{s^2\|z_n\|_2^2/\varrho'}e^{s(\langle \bar{\boldsymbol{\theta}}, z_n\rangle - \delta_1)}] \qquad\qquad \text{Assumption (H)}$$

$$\leq 0.5\mathbb{E}[e^{2s^2\|z_n\|_2^2/\varrho'} + e^{2s(\langle \bar{\boldsymbol{\theta}}, z_n\rangle - \delta_1)}] \qquad\qquad \text{AM-GM inequality}$$

$$\leq 0.5[e^{4s^4\sigma_2^2/\varrho^2 + 2s^2\delta_2/\varrho} + e^{2s^2\sigma_1^2}] \qquad\qquad \text{subgaussianity}$$

$$\leq 0.5[e^{2s^2(1+\delta_2)/\varrho} + e^{2s^2\sigma_1^2}] \qquad\qquad \text{bound on } |s|$$

$$\leq e^{s^2\lambda^2/2}.$$

$\qquad\qquad\qquad\qquad\qquad\qquad\qquad\qquad\qquad\qquad\qquad\qquad\qquad\qquad\qquad\qquad\qquad\qquad \square$

**Corollary C.5.** *Let $\phi_2$ be the second-order Chebyshev approximation to $\phi_{\mathrm{logit}}$ on $[-R, R]$ and let $\tilde{\pi}_{\mathcal{D}}(\boldsymbol{\theta}) = \mathcal{N}(\boldsymbol{\theta} \mid \tilde{\boldsymbol{\theta}}_{MAP}, \tilde{\boldsymbol{\Sigma}})$ denote the posterior approximation obtained by using $\phi_2$ with a Gaussian prior $\pi_0(\boldsymbol{\theta}) = \mathcal{N}(\boldsymbol{\theta} \mid \boldsymbol{\theta}_0, \boldsymbol{\Sigma}_0)$. Let $\bar{\boldsymbol{\theta}} := \int \boldsymbol{\theta}\pi_{\mathcal{D}}(\mathrm{d}\boldsymbol{\theta})$, let $\delta_1 := N^{-1}\sum_{n=1}^N \langle y_n\mathbf{x}_n, \bar{\boldsymbol{\theta}}\rangle$, and let $\sigma_1$ be the subgaussanity constant of the random variable $\langle y_n\mathbf{x}_n, \bar{\boldsymbol{\theta}}\rangle - \delta_1$, where $n \sim \mathsf{Unif}\{1, \ldots, N\}$. Assume that $|\delta_1| \leq R$, that $\|\tilde{\boldsymbol{\Sigma}}\|_2 \leq cd/N$, and that $\|\mathbf{x}_n\|_2 \leq 1$ for all $n = 1, \ldots, N$. Then with $\sigma_0^2 := \|\boldsymbol{\Sigma}_0\|_2$, we have*

$$d_{\mathcal{W}}(\pi_{\mathcal{D}}, \tilde{\pi}_{\mathcal{D}}) \leq cd\left(a(R) + \sqrt{2}\sigma_0 e^{8(2+\sigma_1^2\sigma_0^{-2}) - \sqrt{2}\frac{R-|\delta_1|}{\sigma_0}}\right),$$

*where $a(R)$ is bounded by*

$$\min_{r \in (1, \pi/R + \sqrt{\pi^2/R^2 + 1})} \left|\log\left(1 + e^{-\frac{1}{2}R(r - r^{-1})i}\right)\right| \frac{(r+1)(9r^2 + 7r + 2)}{r^2(r-1)^4}.$$

*Proof.* Assumption (E) holds by construction. The bound on

$$a(R) := \sup_{s\in[-R,R]} |\phi'_{\mathrm{logit}}(s) - \phi'_2(s)|$$

follows immediately from Corollary B.4 in the case of $M = 2$. Furthermore, since $\phi_2'(s) = b_{1,1} + b_{1,2}s$, for $|s| > R$, the additional error is at most $|b_{1,2}|(|s| - R)$. In the case of a Chebyshev approximation, it is easy to verify that $|b_{1,2}| \leq 0.25$ for all $R$ (since as $R \to 0$, $b_{1,2} \to \phi_{\text{logit}}''(0) = -0.25$ and $-b_{1,2}$ is a decreasing function of $R$). In short, $|\phi_{\text{logit}}'(s) - \phi_2'(s)| \leq a(R) + 0.25 \max(0, |s| - R)$ and therefore, using Assumption (F), we have

$$\|\nabla_{\boldsymbol{\theta}}\phi(\langle y_n \mathbf{x}_n, \boldsymbol{\theta}\rangle) - \nabla_{\boldsymbol{\theta}}\phi_M(\langle y_n \mathbf{x}_n, \boldsymbol{\theta}\rangle)\|_2$$
$$= \|\phi'(\langle y_n \mathbf{x}_n, \boldsymbol{\theta}\rangle)y_n\mathbf{x}_n - \phi_M'(\langle y_n \mathbf{x}_n, \boldsymbol{\theta}\rangle)y_n\mathbf{x}_n\|_2$$
$$\leq a(R) + .25\max(0, |\langle y_n \mathbf{x}_n, \boldsymbol{\theta}\rangle| - R).$$

Hence Assumption (G) holds with $a_n = a(R)$ and $b = 0.25$.

Now, clearly $-\log \pi_{\mathcal{D}}$ is $\sigma_0^{-2}$-strongly convex. Since $\|\mathbf{x}_n\|_2 \leq 1$, conclude that $\delta_2 \leq 1$ and $\sigma_2 \leq 1/2$. To upper bound $\varepsilon$, note that

$$B(R, \delta_1, \lambda, \beta) + B(R, -\delta_1, \lambda, \beta) \leq 2B(R, |\delta_1|, \lambda, \beta)$$

and that $\bar{p}(t, \lambda, \beta) \leq e^{\frac{\lambda^2}{4\beta^2}} e^{-t/\beta}$. Also, $\lambda^2 \leq 4(2\sigma_0^2 + \sigma_1^2)$ and $\beta^2 = \sigma_0^2/2$. Using this upper bound in $B(R, |\delta_1|, \lambda, \beta)$ along with straightforward simplifications yields:

$$2B(R, |\delta_1 a|, \lambda, \beta) \leq 2\beta e^{\frac{\lambda^2}{4\beta^2}} e^{-\frac{R - |\delta_1|}{\beta}} \leq \sqrt{2}\sigma_0 e^{8(2 + \sigma_1^2\sigma_0^{-2})} e^{-\sqrt{2}\cdot\frac{R - |\delta_1|}{\sigma_0}}.$$

The result now follows from Theorem C.3 since $-\log\tilde{\pi}_{\mathcal{D}}$ is $\|\tilde{\boldsymbol{\Sigma}}\|_2^{-1}$-strongly convex and hence by assumption $N/(cd)$-strongly convex. $\square$

**Corollary C.6.** *Let $f_M(s)$ be the order-$M$ Chebyshev approximation to $e^t$ on the interval $[-R, R]$, and let $\tilde{\pi}_{\mathcal{D}}(\boldsymbol{\theta})$ denote the posterior approximation obtained by using the approximation $\log\tilde{p}(y_n \,|\, \mathbf{x}_n, \boldsymbol{\theta}) := y_n\mathbf{x}_n \cdot \boldsymbol{\theta} - f_M(\mathbf{x}_n \cdot \boldsymbol{\theta}) - \log y_n!$ with a log-concave prior on $\Theta = \mathbb{B}_R(\mathbf{0})$. If $\inf_{s \in [-R,R]} f_M''(s) \geq \tilde{\varrho} > 0$ and $\|\mathbf{x}_n\|_2 \leq 1$ for all $n = 1, \ldots, N$, then with $\tau := \|\sum_{n=1}^N \mathbf{x}_n\mathbf{x}_n^\top\|_2$, we have*

$$d_{\mathcal{W}}(\pi_{\mathcal{D}}, \tilde{\pi}_{\mathcal{D}}) \leq \frac{N}{\tilde{\varrho}\tau} \min_{r > 1} e^{\frac{1}{2}R(r + r^{-1})} \frac{(r+1)[M^2 r(r+1) + M(2r^2 + r + 1) + r(r+1)]}{r^M(r-1)^4}.$$

Note that $\inf_{s \in [-R,R]} f_M''(s) \geq \tilde{\varrho} > 0$ holds as long as $M$ is even and sufficiently large.

*Proof.* Since by hypothesis $\inf_{s \in [-R,R]} f_M''(s) \geq \tilde{\varrho} > 0$, the prior is log-concave, and $-\log\tilde{\pi}_{\mathcal{D}}$ is $\tilde{\varrho}\tau$-strongly convex (i.e., Assumption (E) holds). Using Assumption (F), we have

$$\|\nabla_{\boldsymbol{\theta}}\log p(y_n \,|\, \mathbf{x}_n, \boldsymbol{\theta}) - \nabla_{\boldsymbol{\theta}}\log\tilde{p}(y_n \,|\, \mathbf{x}_n, \boldsymbol{\theta})\|_2$$
$$= \|e^{\langle y_n \mathbf{x}_n, \boldsymbol{\theta}\rangle}y_n\mathbf{x}_n - f_M'(\langle y_n \mathbf{x}_n, \boldsymbol{\theta}\rangle)y_n\mathbf{x}_n\|_2$$
$$\leq \sup_{s \in [-R, R]} |e^{-s} - f_M'(s)| =: a(R).$$

which is bounded according to Corollary B.5. Hence Assumption (G) holds with $a_n = a(R)$ and $b = 0$. The result now follows immediately from Theorem C.3. $\square$

## Footnotes

[1]A differentiable function $f : \mathbb{R}^d \to \mathbb{R}$ is $\varrho$-strongly convex if for all $\mathbf{v}, \mathbf{w} \in \mathbb{R}^d$, $f(\mathbf{v}) \geq f(\mathbf{w}) + \langle \nabla f(\mathbf{w}), \mathbf{v} - \mathbf{w} \rangle + (\varrho/2)\|\mathbf{v} - \mathbf{w}\|_2^2$.

[2]An arbitrary function $g : \mathbb{R}^d \to \mathbb{R}$ is strictly quasi-concave if for all $\mathbf{v}, \mathbf{w} \in \mathbb{R}^d$, $\mathbf{v} \neq \mathbf{w}$, and $t \in (0, 1)$, $g(t\mathbf{v} + (1-t)\mathbf{w}) > \min\{g(\mathbf{v}), g(\mathbf{w})\}$.