[Reviews · NeurIPS 2017]

Reviewer 1



The authors propose to use polynomial approximate sufficient statistics for non-exponential family models in the generalised linear models. More specifically the authors use Chebyshev polynomials to approximate the mapping function \phi among other polynomials, since the error with the Chebyshev polynomials is exponentially small with order M when the mapping function is smooth. The authors show the theoretical results in Section 4, in which how the error in the MAP estimate decreases exponentially with order M. The authors also show how the posteriors with and without the polynominal approximation differ in terms of the Wasserstein distance, in Section 4.2. I found it a bit difficult to understand what the Corollary 4.3 an 4.4 mean. Can authors elaborate what these mean in plain English in their rebuttal? Also, how are their theoretical results reflected in the experiments? --- after reading their rebuttal --- Thanks the authors for answering my questions. I keep my rating the same.

Reviewer 2



[EDIT] I have read the rebuttal and I have subsequently increased my score to 7. # Summary of the paper When data are too numerous or streaming, the authors investigate using approximate sufficient statistics for generalized linear models. The crux is a polynomial approximation of the link function. # Summary of the review The paper is well written and the method is clear and simple. Furthermore, it is computationally super-cheap when the degree of the polynomial approximation is low, a claim which is backed by interesting experiments. I like the underlying idea of keeping it simple when the model allows it. My main concern is the theoretical guarantees, which 1) lack clarity, 2) involve strong assumptions that should be discussed in the main text, and 3) are not all applicable to the recommended experimental setting. # Major comments - L47 "constant fraction": some subsampling methods have actually been found to use less than a constant fraction of the data when appropriate control variates are available, see e.g. [Bardenet et al, MCMC for tall data, Arxiv 2015], [Pollock et al., the scalable Langevin [...], Arxiv 2016]. Actually, could your polynomial approximation to the likelihood could be useful as a control variate in these settings as well? Your objection that subsampling requires random data access still seems to hold for these methods, though. - L119 "in large part due to...": this is debatable. I wouldn't say MCMC is expensive 'because of' the normalizing constant. "Approximating this integral": it is not the initial purpose of MCMC to approximate the normalizing constant, this task is typically carried out by an additional level above MCMC (thermodynamic integration, etc.). - Section 4: the theoretical results are particularly hard to interpret for several reasons. First, the assumptions are hidden in the appendix, and some of them are quite strong. The strongest assumptions at least should be part of the main text. Second, in -say- Corollary 4.1, epsilon depends on R, and then L207 there's a necessary condition involving R and epsilon, for which one could wonder whether it is ever satisfied. Overall, the bounds are hard to make intuitive sense of, so they should be either simplified or explained. Third, some results like Corollary 4.2 are "for M sufficiently large", while the applications later typically consider M=2 for computational reasons. This should be discussed. - The computational issues of large d and large M should be investigated further. - Corollary 4.2 considers a robust loss (Huber's) but with a ball assumption on all data points, and a log concave prior on theta. I would typically use Huber's loss in cases where there are potential outlier inner products of x and theta; is that not in contradiction with the assumptions? - Corollary 4.3 Assumption (I) is extremely strong. The appendix mentions that it can be relaxed, I think it should be done explicitely.

Reviewer 3



I found the introduction and motivation for this paper to be quite good. General Linear Models (GLM) are certainly a widely used tools so improvements for large scale data analysis problems is impactful. The basic idea of expanding the link function to provide approximate sufficient statistics seems clear enough. I'm not familiar enough with the full literature on scalable versions of GLM to know how this work compares. A more complete literature review would have helped. The abstract for instance says existing approaches either don't scale or don't provide theoretical guarantees on quality, and then says the method shows "competitive performance" (to what?). After reading the introduction (line 43), the authors say variational Bayes "lacks guarantees on quality" without explaining what they mean. I assume this is also what is being referred to in the abstract where they write "don't provide theoretical guarantees on quality". I assume this refers to the fact that the recognition model or approximate posterior in variational Bayes need not be the true posterior if the variational family isn't expressive enough. This seems to be a fairly weak argument against variational Bayes, given that this method is also using approximate sufficient statistics. Regardless, more some more text dedicated to the problem with methods like SGD and VB would be welcome. On page 3 line 116, I would strike the somewhat silly line that the MAP is a strict generalization of the MLE b/c in some situations they are equal. They are different concepts and they have different transformation properties. Eg. from theta -> theta' = g(theta) the MAP can change, but the MLE is invariant. On page 4 line 138 it mentions the log likelihood isn't a dot product of a statistic and the parameter, hence the argue not in the exponential family. This should be clarified as the Poisson distribution is in the exponential family and the footnote on page 3 mentions that they are using a non-standard notation for the exponential family as they have suppressed the base measure and log-partition function. I have a hard time reading Figure 2 given the colored labels. It's also a little strange that some are curves vs. time and the others are simply dots. Also, what should the reader make of the fact the -LL for PASS-LR2 is less than that of the true posterior ? It seems like this is a measure of uncertainty or variance in that estimate. The NIPS review guidelines say: "We would prefer to see solid, technical papers that explore new territory or point out new directions for research, as opposed to ones that may advance the state of the art, but only incrementally. A solid paper that heads in a new and interesting direction is taking a risk that we want to reward." Overall my impression is this is an incremental advance the state of the art.